# Genetic dissection of the different roles of hypothalamic kisspeptin neurons in regulating female reproduction

Luhong Wang[1], Charlotte Vanacker[1], Laura L Burger[1], Tammy Barnes[2], Yatrik M Shah[1], Martin G Myers[1,2], Suzanne M Moenter[2,3]*

[1]Department of Molecular and Integrative Physiology, University of Michigan, Ann Arbor, United States; [2]Department of Internal Medicine, University of Michigan, Ann Arbor, United States; [3]Department of Obstetrics & Gynecology, University of Michigan, Ann Arbor, United States

**Abstract** The brain regulates fertility through gonadotropin-releasing hormone (GnRH) neurons. Estradiol induces negative feedback on pulsatile GnRH/luteinizing hormone (LH) release and positive feedback generating preovulatory GnRH/LH surges. Negative and positive feedbacks are postulated to be mediated by kisspeptin neurons in arcuate and anteroventral periventricular (AVPV) nuclei, respectively. Kisspeptin-specific ERα knockout mice exhibit disrupted LH pulses and surges. This knockout approach is neither location-specific nor temporally controlled. We utilized CRISPR-Cas9 to disrupt ERα in adulthood. Mice with ERα disruption in AVPV kisspeptin neurons have typical reproductive cycles but blunted LH surges, associated with decreased excitability of these neurons. Mice with ERα knocked down in arcuate kisspeptin neurons showed disrupted cyclicity, associated with increased glutamatergic transmission to these neurons. These observations suggest that activational effects of estradiol regulate surge generation and maintain cyclicity through AVPV and arcuate kisspeptin neurons, respectively, independent from its role in the development of hypothalamic kisspeptin neurons or puberty onset.
DOI: https://doi.org/10.7554/eLife.43999.001

*For correspondence:
smoenter@umich.edu

Competing interests: The authors declare that no competing interests exist.

## Introduction

Infertility is a common clinical problem affecting 15% of couples; ovulatory disorders account for 25% of this total (*Macaluso et al., 2010*). The hypothalamic-pituitary-gonadal axis controls reproduction and malfunction of this axis can cause ovulatory dysfunction and/or other disturbances of the reproductive cycle (*Helm et al., 2009*; *Plant and Zelesnik, 2015*). Gonadotropin-releasing hormone (GnRH) neurons form the final common pathway for central neural regulation of reproduction. GnRH stimulates the pituitary to secrete follicle-stimulating hormone and luteinizing hormone (LH), which regulate gonadal steroid and gamete production. Estradiol, via estrogen receptor alpha (ERα), plays crucial roles in both homeostatic negative feedback and positive feedback action on GnRH/LH release in females (*Döcke and Dörner, 1965*; *Moenter et al., 1990*; *Lubahn et al., 1993*; *Krege et al., 1998*; *Wintermantel et al., 2006*; *Christian et al., 2008*; *Glanowska et al., 2012*; *Cheong et al., 2015*). Low estradiol levels suppress pulsatile GnRH/LH release, whereas sustained elevations in estradiol during the late follicular phase of the cycle cause a switch of estradiol feedback action from negative to positive, inducing prolonged GnRH/LH surges, which ultimately triggers ovulation (*Christian and Moenter, 2010*). As GnRH neurons typically do not express detectable ERα (*Hrabovszky et al., 2001*), estradiol feedback is likely transmitted to GnRH neurons by ERα-expressing afferents.

**eLife digest** Female reproduction relies on a complex balance of hormones that drive the reproductive cycle (menstrual cycle in humans) and influence fertility. A hormone called GnRH, which stands for gonadotropin-releasing hormone, plays a major role in regulating this balance. GnRH is transmitted from the brain and stimulates the release of other hormones from a nearby gland called the pituitary gland, which, in turn, activates the reproductive organs to produce steroid hormones, such as estrogen.

Steroids do many things in the body, including regulating the release of GnRH and pituitary hormones through a process called feedback. In the case of negative feedback, steroids maintain the release of GnRH and pituitary hormone within a normal range. Once per reproductive cycle, estrogen will instead positively feed back into the system and activate GnRH, causing pituitary hormone levels to spike, and initiate the release of one or more eggs from the ovary by a process known as ovulation. The neurons that make GnRH do not directly respond to estrogen, but instead receive input from different upstream neurons that contain estrogen receptors. However, it is poorly understood how fertility is regulated by these neurons.

To investigate the effects of estrogen on these upstream neurons, Wang et al. genetically removed the estrogen receptors from two separate populations of neurons in mice. Estrogen was found to affect each of these populations differently, inhibiting one and activating the other. Wang et al. showed that these two populations likely have different roles in reproduction: the population inhibited by estrogen regulates negative feedback and generates reproductive cycles, whilst the population activated by estrogen regulates positive feedback and stimulates ovulation.

This knowledge furthers our understanding of how the brain regulates fertility, and the genetic approach used to remove the estrogen receptor could be applied to the study of other hormones that act on the brain.

DOI: https://doi.org/10.7554/eLife.43999.002

Kisspeptin neurons in the arcuate and anteroventral periventricular (AVPV) regions are estradiol-sensitive GnRH afferents that are postulated to mediate estradiol negative and positive feedback, respectively (*Oakley et al., 2009*; *Lehman et al., 2010*). Kisspeptin potently stimulates GnRH neurons and *Kiss1* mRNA is differentially regulated in these nuclei by estradiol (*Han et al., 2005*; *Messager et al., 2005*; *Smith et al., 2005*; *Pielecka-Fortuna et al., 2008*; *Lehman et al., 2010*; *Kumar et al., 2015*; *Yip et al., 2015*). ERα in kisspeptin cells is critical for estradiol negative and positive feedback, as kisspeptin-specific ERα knockout (KERKO) mice exhibit higher frequency LH pulses and fail to exhibit estradiol-induced LH surges (*Mayer et al., 2010*; *Dubois et al., 2015*; *Greenwald-Yarnell et al., 2016*; *Wang et al., 2018*). Although informative, the KERKO model has several caveats that limit interpretation. First, ERα is deleted as soon as *Kiss1* is expressed, before birth in arcuate kisspeptin neurons (also called KNDy neurons for coexpression of kisspeptin, neurokinin B and dynorphin) and before puberty in AVPV kisspeptin neurons (*Semaan et al., 2010*; *Kumar et al., 2014*). This may cause developmental changes in these cells and/or their networks. Second, ERα is deleted from all kisspeptin cells, thus making it impossible to assess independently the role of AVPV and arcuate kisspeptin neurons.

Combining CRISPR-Cas9 with targeted viral vector injection allows deletion of ERα in a nucleus-specific and temporally-controlled manner to address the above caveats (*Swiech et al., 2015*). We designed Cre-dependent AAV vectors that carry single guide RNAs (sgRNAs) that target *Esr1* (encoding ERα) or *lacZ* and delivered these vectors to the AVPV or arcuate of adult female mice that express Cas9 in kisspeptin cells. We then compared the reproductive phenotypes as well as kisspeptin neuronal physiology in AAV-*Esr1* vs AAV-*lacZ* targeted mice and KERKO mice.

## Results

### AVPV kisspeptin neurons from KERKO mice exhibit decreased firing rate and excitability

We first used extracellular recordings to monitor the spontaneous firing rate of YFP-identified AVPV kisspeptin neurons in coronal brain slices from ovary-intact control and KERKO mice. As the persistent cornified vaginal cytology of KERKO mice is similar to that observed during estrus (*Greenwald-Yarnell et al., 2016*), we used mice in the estrous stage of the reproductive cycle as controls. The firing frequency of AVPV kisspeptin neurons was lower in ovary-intact KERKO mice compared to controls (*Figure 1a,b*, two-way ANOVA/Holm-Sidak, p=0.0001). To test if the firing rate of AVPV kisspeptin neurons in KERKO mice responds to circulating estradiol, we repeated this study in ovariectomized (OVX) mice and OVX mice with an estradiol implant producing constant physiologic levels (OVX + E) (*Christian et al., 2005*). OVX reduced and estradiol treatment increased firing rate in cells from control, but not KERKO, mice (*Figure 1a,b* two-way ANOVA/Holm-Sidak, control intact vs OVX p=0.009, intact vs OVX + E, p=0.02, OVX vs OVX + E, p<0.0001). As a result of this difference, the firing frequency is higher in cells from OVX + E control than OVX + E KERKO mice (p<0.0001). Statistical test parameters for all figures are in *Tables 1* and *2*.

We next recorded the whole-cell firing signatures of neurons in these six groups in response to current injection. AVPV kisspeptin neurons in control mice mice exhibit a greater number of depolarization-induced bursts (DIB) and rebound bursts when estradiol is elevated, confirming previous observations (*Wang et al., 2016*) (*Figure 1c,d,e*, Chi-square, DIB, p=0.02; rebound, p=0.02; Fisher's exact *post hoc* test, DIB, OVX vs OVX +E, p=0.008, rebound OVX vs OVX +E p=0.03, for other paired comparisons, p>0.2). In KERKO mice, these two types of bursts were rare (<25% of cells) in all steroid conditions tested and were not regulated by estradiol (*Figure 1c,d,e*, Chi-square, DIB, p=0.4, rebound, p=0.3). We also compared the action potential output of these cells in response to current injection (0–50 pA, 10 pA increments, 500 ms). Cells from ovary-intact KERKO mice generated fewer action potentials compared to controls. Action potential generation as a function of current injection was similar in cells from OVX control and OVX KERKO mice but was increased by estradiol only in control mice (*Figure 1f*, two-way repeated-measures ANOVA/Holm-Sidak, intact, 20 pA, p=0.03, 30 pA, p=0.06; OVX +E, 20 pA to 50 pA, p≤0.04). Reduced action potential firing of AVPV kisspeptin neurons from KERKO mice may be attributable at least in part to decreased input resistance compared to controls (*Figure 1—figure supplement 1*, two-way ANOVA/Holm-Sidak control vs KERKO, intact, p=0.006, OVX, p=0.7, OVX +E p=0.02).

As both depolarization-induced bursts and rebound bursts are sensitive to NiCl (100 µM) (*Lee et al., 1999*) at levels that fairly specifically block T-type calcium channels, we measured T-type ($I_T$) current density and voltage dependence. $I_T$ current density was decreased in AVPV kisspeptin cells from gonad-intact KERKO mice compared to controls (*Figure 2a,b*, two-way repeated-measures ANOVA/Holm-Sidak, −50 mV, p=0.003; −40 mV, p=0.002; −30 mV, p=0.003). The voltage dependence of activation was not different between groups, but the voltage dependence of inactivation was depolarized in cells from KERKO mice (*Figure 2c*, control vs KERKO, two-tailed unpaired Student's *t*-test, $V_{1/2}$ activation −52.2 ± 1.6 vs −48.6 ± 1.4 mV, p>0.1; slope 5.5 ± 0.6 vs 5.5 ± 0.7, p>0.1; $V_{1/2}$ inactivation −74.8 ± 4.1 vs −61.9 ± 3.1 mV, p=0.03; slope −3.1 ± 0.5 vs −4.2 ± 0.3, p=0.1).

### Design and validation of sgRNAs that target *Esr1*

A caveat of studying the role of ERα in AVPV kisspeptin neurons using KERKO mice is that the deletion of ERα (encoded by *Esr1*) using cre recombinase under the control of the kisspeptin promoter is neither time- nor location-specific. We utilized the CRIPSR-Cas9 approach to achieve temporal and spatial control of *Esr1* gene knockdown. Two sgRNAs were designed that target exon1 of *Esr1* based on software prediction (*Ran et al., 2013*); sites predicted by FengZhang's guide design software (http://crispr.mit.edu) as possible off-target regions for binding of these guides are listed in *Table 3*. The efficiency of each guide was tested in vitro in C2C12 mouse myoblast cells (*Milanesi et al., 2008*). The sgRNAs that target *Esr1* and a sgRNA that targets *lacZ* as a control were subcloned into the lentiCRISPRv2 plasmid (*Sanjana et al., 2014*), from which Cas9 and the sgRNA are expressed after transfection of C2C12 cells. Puromycin was used to select construct-

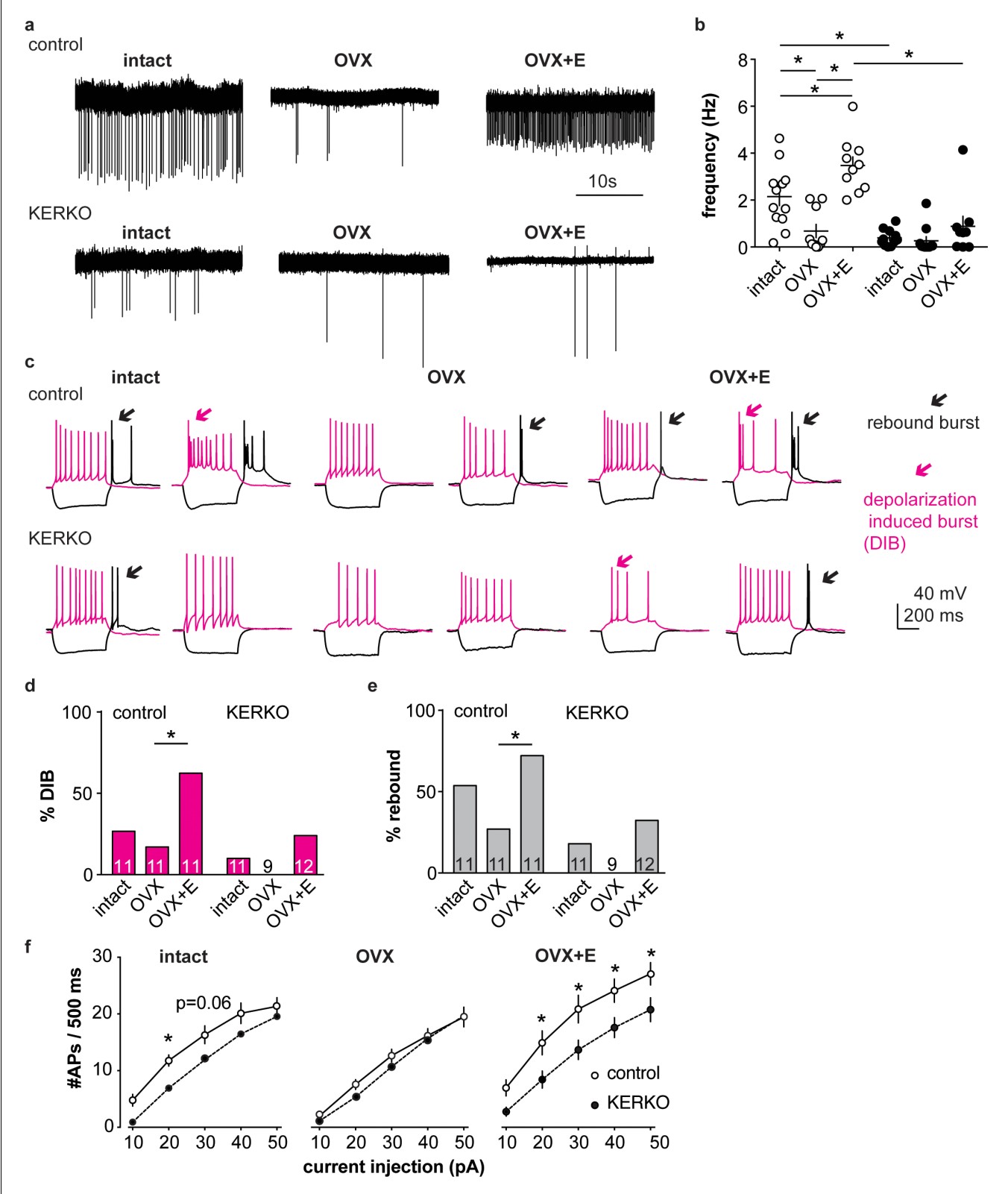

**Figure 1.** AVPV kisspeptin neurons from KERKO mice are less excitable compared to those from control mice and are not regulated by estradiol. (**a**) Representative extracellular recordings for cells from control and KERKO mice from ovary-intact, OVX and OVX +E groups. (**b**) Individual values and mean ± SEM firing frequency of cells from control (white circles) and KERKO groups (black circles). (**c**) representative depolarizing (magenta,+20 pA, 500 ms) and hyperpolarizing (black, −20 pA, 500 ms) firing signatures for cells from control and KERKO mice in ovary-intact (left), OVX (middle) and OVX +E

*Figure 1 continued on next page*

*Figure 1 continued*

(right) groups; black arrows indicate rebound bursts and red arrows indicate depolarization-induced bursts (DIB). Initial membrane potential was 70 ± 2 mV. (**d**) and (**e**) percent of cells exhibiting DIB (**d**) or rebound (**e**) bursts; cells per group is shown within the bar. (**f**) Input-output curves for cells from control and KERKO mice; ovary-intact (left), OVX (middle) and OVX +E (right). *p<0.05.

DOI: https://doi.org/10.7554/eLife.43999.003

The following figure supplement is available for figure 1:

**Figure supplement 1.** Recording parameters and uterine mass.

DOI: https://doi.org/10.7554/eLife.43999.004

expressing cells. After a ~ 4-week selection period, DNA was harvested and the *Esr1* region sequenced. Cells expressing either of the sgRNAs targeting *Esr1,* but not *lacZ,* exhibited a peak-on-peak sequencing pattern, indicating disruption of the gene (*Figure 3a*). As these in vitro experiments suggested these sgRNAs were able to mutate *Esr1,* we designed Cre-dependent AAV vectors to express each sgRNA and mCherry (to indicate infected cells) under control of the U6 promoter (*Figure 3b*). The AAV vector was bilaterally stereotaxically injected into the AVPV region of adult female mice that express Cas9 and GFP under control of the kisspeptin promoter (*Kiss1*-Cre; *Cas9 loxp*-stop-*Gfp*, *Figure 3—figure supplement 1*); these groups are referred to as AVPV-AAV-*Esr1* or AVPV-AAV-*lacZ*. Only one guide was injected per animal to allow comparison of phenotypes when different areas of *Esr1* were targeted. The ERα knockdown efficiency of the two sgRNAs target *Esr1* was comparable. The infection rate for AVPV-AAV-*Esr1* was 81 ± 4% (*Figure 3d*, *Esr1*-guide 1 [g1] 82 ± 2%, n = 3; *Esr1*-guide 2 [g2] 81 ± 8%, n = 3) and only 28 ± 1% of AVPV kisspeptin cells expressed ERα post infection (*Figure 3d*, n = 3, *Esr1*-guide1 27 ± 0.4%; n = 3, *Esr1*-guide2 28 ± 2%). In mice that received AVPV-AAV-*lacZ* (n = 3), the infection rate was comparable at 82 ± 2%, but

**Table 1.** Statistical parameters for two-way ANOVA.

| Parameter | Figure | Factor 1 | Factor 2 | Interaction |
|---|---|---|---|---|
| Firing frequency | *Figure 1b* | steroid $F_{(2, 57)}=14.7$* | genotype $F_{(1, 57)}=40.1$* | $F_{(2, 57)}=6.2$* |
| Input-output curve | *Figure 1f* intact OVX OVX + E | current $F_{(4, 80)}=242.7$* $F_{(4, 72)}=138.6$* $F_{(4, 84)}=182.2$* | genotype $F_{(1, 20)}=8.2$* $F_{(1, 18)}=0.8$ $F_{(1, 21)}=6.6$* | $F_{(4, 80)}=0.6$ $F_{(4, 72)}=0.7$ $F_{(4, 84)}=1.5$ |
| $I_T$ current density | *Figure 2b* | voltage $F_{(8, 104)}=39.74$* | genotype $F_{(1, 13)}=11.1$* | $F_{(8, 104)}=9.4$* |
| $I_T$ normalized conductance | *Figure 2c* activation inactivation | voltage $F_{(8, 104)}=494.7$* $F_{(8, 104)}=195.8$* | genotype $F_{(1, 13)}=3.2$ $F_{(1, 13)}=4.5$* | $F_{(8, 104)}=1.5$ $F_{(8, 104)}=3.1$* |
| LH | *Figure 3f* *Figure 3g* | AAV type $F_{(1, 12)}=29.8$* $F_{(1, 13)}=0.3$ | time $F_{(2, 24)}=2.1$ $F_{(1, 13)}=35.8$* | $F_{(2, 24)}=1.8$ $F_{(1, 13)}=19.5$* |
| Input-output curve | *Figure 4d* IF post hoc PCR post hoc AAV-*Esr1* vs KERKO | current $F_{(4, 136)}=165.5$* $F_{(4, 68)}=123$* $F_{(4, 100)}=154.7$* | AAV type $F_{(2, 34)}=7.2$* $F_{(1, 17)}=12.5$* $F_{(1, 25)}=2.1$ | $F_{(8, 136)}=0.7$ $F_{(4, 68)}=4.3$* $F_{(4, 100)}=7.2$ |
| Days proestrus/week | *Figure 5d* | time $F_{(1, 12)}=13.6$* | AAV type $F_{(2, 12)}=5.8$* | $F_{(2, 12)}=10.0$* |
| LH | *Figure 5g* kisspeptin GnRH | injection $F_{(1, 12)}=34.8$* $F_{(1, 12)}=20.0$* | AAV type $F_{(1, 12)}=4.7$ # $F_{(1, 12)}=7.0$* | $F_{(1, 12)}=17.1$* $F_{(1, 12)}=7.5$* |
| | | steroids | genotype | interaction |
| Input resistance | *Figure 1—figure supplement 1a* | $F_{(2, 59)}=2.6$ | $F_{(1, 59)}=13.2$* | $F_{(2, 59)}=2.0$ |
| Cell capacitance | *Figure 1—figure supplement 1b* | $F_{(2, 59)}=5.2$ | $F_{(1, 59)}=0.1$* | $F_{(2, 59)}=0.4$ |
| Normalized uterine mass | *Figure 1—figure supplement 1c* | $F_{(2, 30)}=19.9$* | $F_{(1, 30)}=80.0$* | $F_{(2, 30)}=2.4$ |

*$p<0.05$, # $p=0.05$

DOI: https://doi.org/10.7554/eLife.43999.005

**Table 2.** Statistical parameters for two group comparisons.
For normally distributed data, two-tailed unpaired Student's t-test; for non-normally distributed data, two-tailed Mann-Whitney U test.

| Parameter | Figure | T or U, df |
|---|---|---|
| $V_{1/2}$ activation slope<br>$V_{1/2}$ inactivation slope inactivation | in the text, control vs KERKO<br>$I_T$ kinetics | t = 1.7, 13<br>t = 0.01, 13<br>t = 2.5, 13<br>t = 1.6, 13 |
| rate of rise IF rate of rise PCR<br>FWHM IF<br>FWHM PCR<br>AHP amplitude IF<br>AHP amplitude PCR | *Figure 4h*<br>*Figure 4i*<br>*Figure 4j* | t = 2.5, 27<br>t = 2.7, 17<br>U = 62 t = 3.1, 17<br>t = 4.4, 27<br>t = 2.7, 27 |
| LH pulses/h | *Figure 5e* | t = 1.7, 12 |
| Mean LH | *Figure 5f* | t = 0.05, 12 |
| Firing rate | *Figure 6b* | U = 45.5 |
| EPSC frequency | *Figure 6e* | t = 4.0, 20 |
| EPSC amplitude | *Figure 6e* | t = 2.7, 20 |
| Input resistance lacZ vs Esr1 IF, lacZ vs Esr1 PCR,<br>KERKO vs Esr1 | *Figure 1—figure supplement 1a* | t = 0.7, 27<br>t = 1.0, 17<br>t = 1.8, 35 |
| Cell capacitance lacZ vs Esr1 IF, lacZ vs Esr1 PCR,<br>KERKO vs Esr1 | *Figure 1—figure supplement 1b* | t = 0.4, 27<br>t = 0.3, 17<br>t = 0.3, 35 |
| Normalized uterine mass lacZ vs Esr1 AVPV lacZ vs Esr1 arcuate | *Figure 1—figure supplement 1c* | t = 0.5, 14<br>t = 2.9, 8 |

*p<0.05, # p=0.05

DOI: https://doi.org/10.7554/eLife.43999.006

there was no disruption of ERα; 72 ± 2% of AVPV kisspeptin neurons expressed ERα, which is similar to control mice (*Kumar et al., 2015*). Of note, the ERα antibody used recognizes the C-terminus, suggesting a lack of rare splice variants that were generated at low levels in initial ERKO mice (*Couse et al., 1995*).

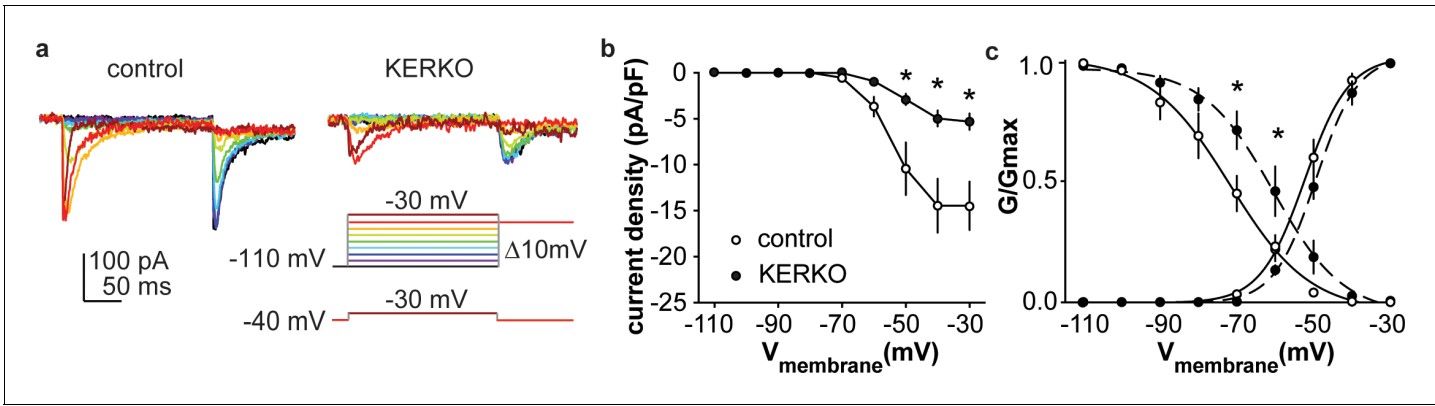

**Figure 2.** T-type calcium currents are reduced in AVPV kisspeptin neurons from KERKO compared to control mice. (a) Voltage protocol (bottom right) and representative $I_T$ in control (left) and KERKO groups (right). (b) Mean ± SEM $I_T$ current density in control (white symbols) and KERKO groups (black symbols). (c) Voltage dependence of $I_T$ conductance activation and inactivation in cells from control and KERKO mice. *p<0.05.

DOI: https://doi.org/10.7554/eLife.43999.007

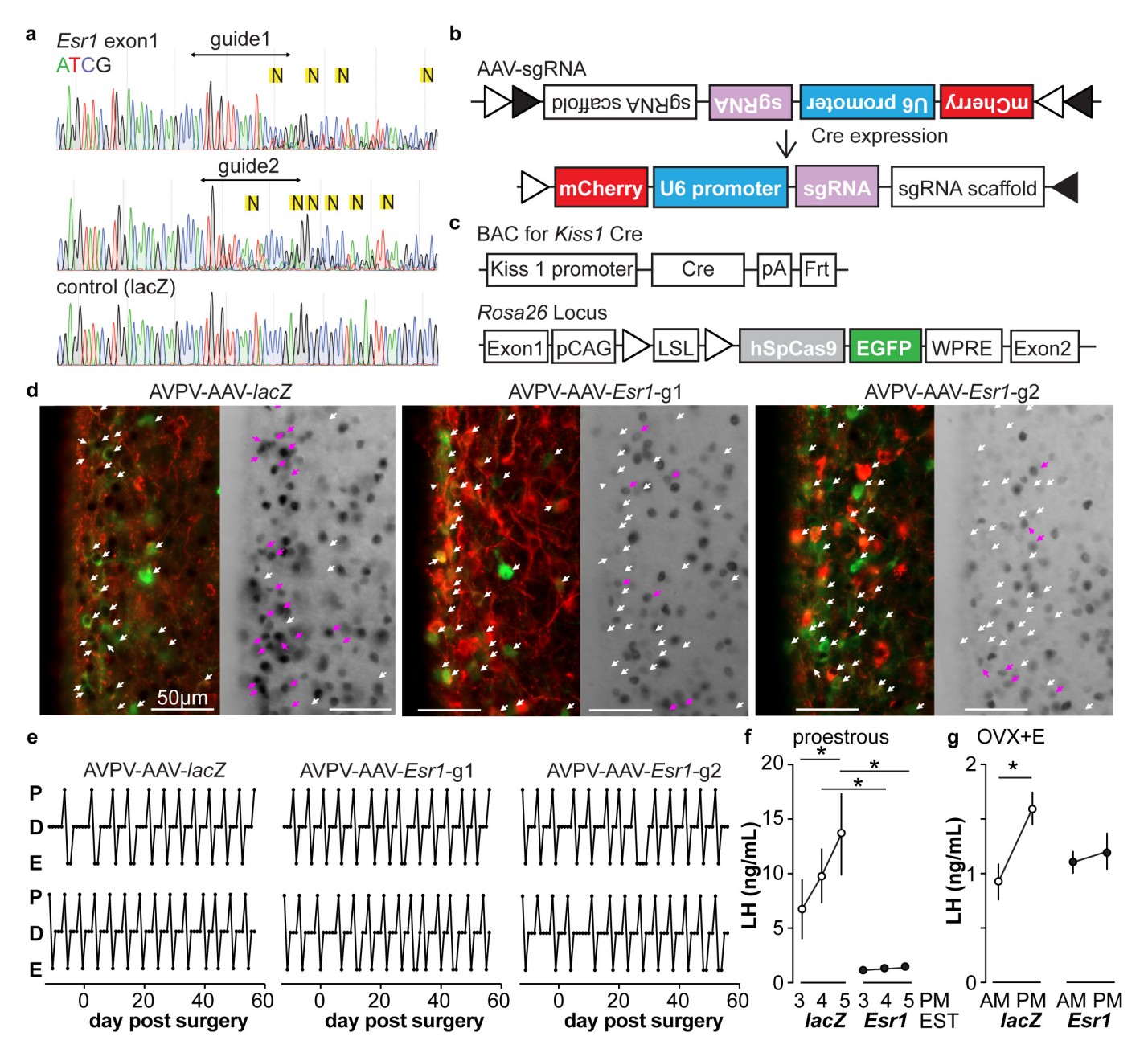

**Figure 3.** In vitro and in vivo validation of AVPV-AAV-*Esr1* guides. (a) Sequencing from C2C12 cells transiently transfected with lentiCRISPR v2 with sgRNAs targeting *Esr1* (guide 1 [g1] top, guide 2 [g2] middle) or lacZ. N in yellow highlight indicates peak on peak mutations. (b) and (c) Schematic representation of (b) the Cre-inducible AAV vector delivering sgRNAs and (c) *Kiss1*-cre *Cas9-loxp Stop-Gfp* mice. (d) AVPV-AAV-*lacZ*, -*Esr1* g1 or g2 were bilaterally delivered to the AVPV region (see *Figure 3—figure supplement 1*). Brain sections were processed to detect GFP (green), mCherry (red) and ERα (black), dual GFP/mCherry detection indicates infection of kisspeptin neuron (white arrows, left panel of each pair). AVPV-AAV-*Esr1* infected AVPV kisspeptin neurons exhibit decreased ERα expression compared to AVPV-AAV-*lacZ* infected cells (right panel of each pair, white arrows indicate ERα-negative, magenta arrows indicate ERα-positive infected cells). (e) Representative reproductive cycles of mice that received AAV-*lacZ, g1* or g2; E, estrus; D, diestrus; P, proestrus; day 0 is the day of stereotaxic surgery. (f) Mean ± SEM proestrous LH surge measured at 3, 4, and 5 pm EST in AVPV-AAV-*lacZ* and AVPV-AAV-*Esr1* mice (mice receiving g1 or g2 combined). (g) Mean ± SEM estradiol-induced LH surge measured at 9 am and 5 pm EST from AAV-*lacZ* and AAV-*Esr1* OVX + E mice (mice receiving g1 or g2 were combined).

DOI: https://doi.org/10.7554/eLife.43999.008

The following figure supplement is available for figure 3:

**Figure supplement 1.** Bilateral delivery of AAV-*lacZ*, and AAV-*Esr1* (g1 and g2) to AVPV of adult female mice.

*Figure 3 continued on next page*

*Figure 3 continued*

DOI: https://doi.org/10.7554/eLife.43999.009

## Knockdown of ERα in AVPV kisspeptin neurons in adulthood does not affect estrous cycles but disrupts preovulatory and estradiol-induced LH surges

We monitored the reproductive cycles of the mice injected with AAV-sgRNAs in the AVPV 12 days before and for up to eight weeks following surgery. Neither AVPV-AAV-*Esr1* guide (tested independently) nor the AVPV-AAV-*lacZ* disrupted reproductive cyclicity (*Figure 3e*), even in mice with a high rate of bilateral infection (~80%). These mice entered proestrus at the same frequency in the last four weeks compared to the first four weeks (two weeks pre-surgery plus the first two weeks post-surgery, two-way repeated-measures ANOVA/Holm-Sidak, before vs after; g1, n = 3, 1.3 ± 0.1 vs 1.6 ± 0.1; g2, n = 4, 1.2 ± 0.1 1.4 ± 0.1; *lacZ* n = 4, 1.3 ± 0.2 vs 1.3±0.1, p>0.1 for each paired comparison). To test for the occurrence of estradiol-positive feedback, we monitored both proestrous (preovulatory) and estradiol-induced LH surges in these mice. Surge data were similar for guide 1 and guide 2 and data from both guides were combined for group comparisons. Both proestrous and estradiol-induced LH surges were blunted after ERα knockdown (*Figure 3f,g*, two-way repeated-measures ANOVA/Holm-Sidak; f, *lacZ*, 3pm vs 5pm, p=0.04; *lacZ* vs *Esr1*, 4pm, p=0.006, 5pm, p<0.0001, h, *lacZ* AM vs PM, p<0.0001). There were fewer corpra lutea (CL) in mice with *Esr1* guides targeted to the AVPV (p<0.05, guides 1 and 2 combined n = 6, 5.2 ± 2.1 CL/mouse, vs *lacZ* guide n = 5, 10.8 ± 0.4 CL/mouse, two-tailed paired Student's t test with Welch's correction, t = 2.598, df = 5.305). Of note, variation was high in the *Esr1* mice, with two looking similar to controls, two having fewer CL and two not having any CL. This suggests ovulation is disrupted in a substantial subpopulation of these mice but can proceed with the blunted LH surge in some animals.

## Decreased excitability of AVPV kisspeptin neurons in AAV-*Esr1* knockdown mice

To test if knockdown of ERα in adult AVPV kisspeptin neurons alters their intrinsic excitability, we recorded firing signatures of infected and uninfected cells in brain slices from AAV injected OVX + E mice. We again observed no difference between AVPV-AAV-*Esr1* g1 vs g2 and combined these data. Some cells were loaded with neurobiotin during recording for identification and ERα protein detected *post hoc* with immunofluorescence (*Figure 4a,c*, and IF post hoc portions of *Figure 4e–j*). Cells not infected with AVPV-AAV-*Esr1* and cells infected with either AVPV-AAV-*Esr1* guide but in which ERα protein was detected exhibited similar firing signatures in terms of DIB and rebound bursts (*Figure 4b,e,f*). In contrast, cells infected by AVPV-AAV-*Esr1* that had undetectable ERα protein had reduced burst firing compared to AVPV-AAV-*lacZ* or uninfected groups (*Figure 4e,f*, Chi-square, DIB, p=0.008, rebound bursts, p=0.0008; Fisher's exact *post hoc* test, DIB, *Esr1* vs *lacZ*, or vs uninfected, p≤0.03; rebound, *Esr1* vs *lacZ*, or vs uninfected p≤0.04; for other paired comparisons, p>0.5). The firing signature of AAV-*Esr1*-infected cells with successful deletion of ERα was comparable to cells from KERKO mice (Chi-square, p>0.9 for both DIB and rebound bursts). Cells that lost detectable ERα after AVPV-AAV-*Esr1* infection also produced fewer action potentials with current injection than cells infected with AAV-*lacZ* (*Figure 4d* left and center, two-way repeated-measures ANOVA/Holm-Sidak, *Esr1* vs *lacZ*, 20 pA, p=0.08; 30 to 50 pA p≤0.02). This difference is not attributable to passive properties (*Figure 1—figure supplement 1a,b*). The relationship between current injection and number of action potentials fired (input-output curve) in cells from KERKO and in AVPV-AAV-*Esr1* knockdown mice was only different at the highest level of current injected, with AVPV-AAV-*Esr1*-infected cells being less excitable (*Figure 4d*, right, two-way repeated-measures ANOVA/Holm-Sidak, 50 pA, p=0.01), despite no change in input resistance (*Figure 1—figure supplement 1a,b* KERKO vs AAV, p=0.08). Action potential properties from AVPV-AAV-*Esr1* knockdown cells with AVPV-AAV-*lacZ* control also differed. Specifically, loss of ERα led to decreased action potential rate of rise, a trend toward prolonged full-width half-maximum (FWHM), and hyperpolarized afterhyperpolarization potential (AHP) (*Figure 4–j*, two-tailed unpaired Student's *t*-test, h, p=0.02, j, p=0.0002; i, Mann-Whitney U-test, p=0.06).

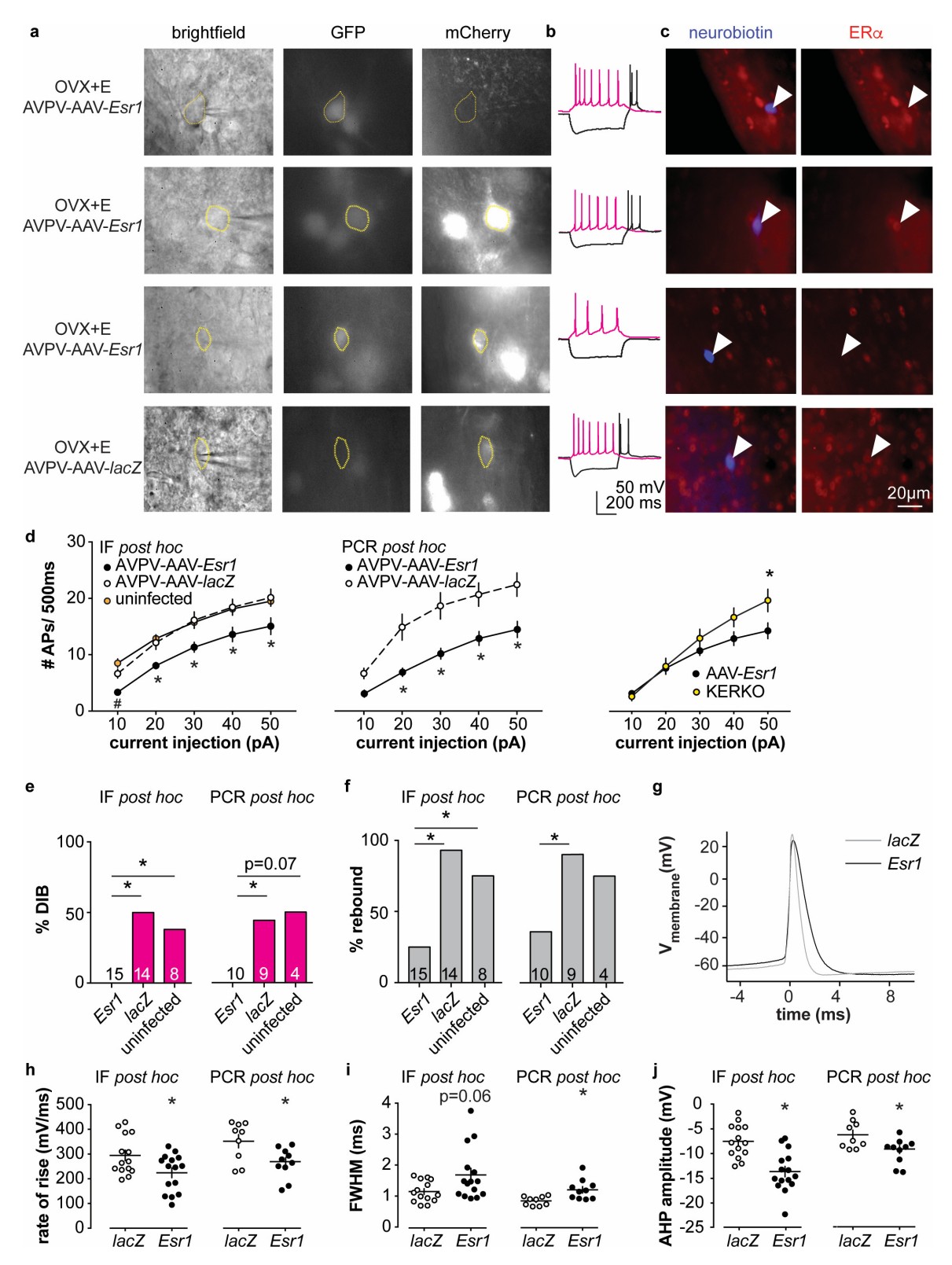

**Figure 4.** Decreased excitability of AVPV kisspeptin neurons in AVPV-AAV-*Esr1* knockdown mice. (a–c) whole-cell recording and immunofluorescence (IF) *post hoc* identification of ERα in recorded cells in OVX + E AVPV-AAV-*Esr1* infected mice. (a) visualization during recording; (b) representative depolarizing (+20 pA, magenta) and hyperpolarizing (−20 pA, black) firing signatures. (c) neurobiotin (blue) and ERα (red) staining after photobleaching of GFP and mCherry signals. From top to bottom: cells not infected by AVPV-AAV-*Esr1* and immunopositive for ERα; cells infected by AVPV-AAV-*Esr1*

*Figure 4 continued on next page*

*Figure 4 continued*

but still immunopositive for ERα; cells infected by AAV-*Esr1* and not immunopositive for ERα; cells infected by AVPV-AAV-*LacZ* and immunopositive for ERα. (d) left, input-output curves of infected cells with undetectable ERα in AAV-*Esr1* (third row in a-c, black circle), cells infected by *AVPV-AAV-lacZ* (bottom row in a-c, white circle, n = 14), and cells not infected by AAV (top row in a-c, orange circle); middle, input-output curves from a separate set of cells in which *Esr1* status was confirmed by single-cell qPCR *post hoc* (AAV-*Esr1* black circle; AAV-*lacZ*, white circle); right, input-output curve of AVPV-AAV-*Esr1* knockdown (black circle) vs KERKO (yellow circle) cells. (e,f) percent of cells exhibiting DIB (e) or rebound bursts (f). Cells per group is shown within or on top of the bar. (g), representative action potentials at the rheobase from *lacZ* vs *Esr1* infected cells. (h–j) individual values and mean ± SEM rate of rise (h) full width at half maximum (FWHM) (i) and afterhyperpolarization potential (AHP) amplitude (j). *p<0.05 vs all other groups; # p<0.05 vs uninfected.

DOI: https://doi.org/10.7554/eLife.43999.011

The following figure supplement is available for figure 4:

**Figure supplement 1.** Single-cell qPCR for mRNA from AVPV kisspeptin neurons in mice with AAV vector delivered to AVPV region.

DOI: https://doi.org/10.7554/eLife.43999.012

In parallel, we performed whole-cell patch-clamp recording with single-cell PCR post hoc identification of *Esr1* mRNA on a separate set of cells (AVPV-AAV-*Esr1*, 10 cells from four mice; AVPV-AAV-*lacZ*, 9 cells from three mice; primers are in *Table 4*). A similar decrease in burst firing and action potential input-output curve was observed in *Esr1* mRNA negative cells as was observed in cells verified to have undetectable ERα protein by immunofluorescence (*Figure 4e,f*, Chi-square, DIB, p=0.04, rebound p=0.03; Fisher's exact *post hoc* test, DIB, *Esr1* vs *lacZ* p=0.03. *Esr1* vs uninfected, p=0.07; rebound, *Esr1* vs *lacZ* p=0.02; for other paired comparisons, p>0.2). Absence of *Esr1* mRNA expression was again associated with decreased number of action potentials in response to current injection (*Figure 4d* middle, *Esr1* vs *lacZ*, p<0.002 for 20 to 50 pA steps). Absence of *Esr1* mRNA, similar to loss of ERα protein, led to decreased action potential rate of rise, prolonged FWHM, and AHP (*Figure 4g–j*, two-tailed unpaired Student's *t*-test, h, p=0.02; i, p=0.006; j, p=0.02). Single-cell PCR analysis also indicates that a lower percent of AVPV-AAV-*Esr1* knockdown cells express *Kiss1* and a trend to increase in *Esr2* mRNA (AVPV-AAV-*Esr1*, 23 cells from four mice; AVPV-AAV-*lacZ*, 16 cells from three mice; *Figure 4—figure supplement 1*). Interestingly, expression of the mRNA for progesterone receptor (*Pgr*) did not differ between groups (*Figure 4—figure supplement 1*), suggesting the estradiol-dependence of this gene may be paracrine regulated in the brain as in other tissues (*Hilton et al., 2015*). We also examined gene expression for several ion channels, but none showed any changes or patterns of expression among groups (*Figure 4—figure supplement 1*).

## Knockdown of ERα in arcuate kisspeptin neurons in adulthood disrupts estrous cycles

To examine the role of estradiol feedback on arcuate kisspeptin neurons, we delivered the AAV-sgRNAs bilaterally to the arcuate region to knockdown ERα in these cells (*Figure 5—figure supplement 1*); these groups are referred to as Arc-AAV-*Esr1* or Arc-AAV-*lacZ*. The infection rate for Arc-AAV-*Esr1* was 92 ± 3% (*Figure 5—figure supplement 1*, n = 3 *Esr1*-guide1 96 ± 2%; n = 3 *Esr1*-guide2 86 ± 2%) and only 34 ± 3% of KNDy neurons expressed ERα post infection (*Figure 5a* n = 3 *Esr1*-g1, 38 ± 0.3%; n = 3 *Esr1*-g2, 30 ± 5%). In mice that received Arc-AAV-*lacZ*, the infection rate was comparable (*Figure 5a* n = 3, 94 ± 3%), and 92 ± 1% of KNDy neurons expressed ERα, similar to control mice (*Kumar et al., 2015*). Reproductive cycles were monitored for 12 days before and for up to eight weeks following surgery. In contrast to mice with Arc-AAV-*Esr1* targeted to the AVPV region, mice with the same virus targeted to the arcuate began exhibiting disrupted cyclicity three to four weeks post surgery (*Figure 5b*). These mice entered proestrus less frequently after surgery than before (two weeks pre-surgery plus the first two weeks post-surgery, *Figure 5d*, two-way repeated-measures ANOVA/Holm-Sidak, g1, p=0.002, g2, p=0.03). There was no difference in LH pulse frequency measured on the day of estrus or mean levels between Arc-AAV-*Esr1* vs Arc-AAV-*lacZ* injected mice on estrus (*Figure 5c,e,f*). Notably, LH response to kisspeptin and to GnRH was reduced in Arc-AAV-*Esr1* mice (*Figure 5g*, two-way repeated measures ANOVA/Holm-Sidak, *lacZ*, control vs kisspeptin or GnRH, p<0.001; *Esr1* vs *lacZ* for kisspeptin and GnRH both, p≤0.002).

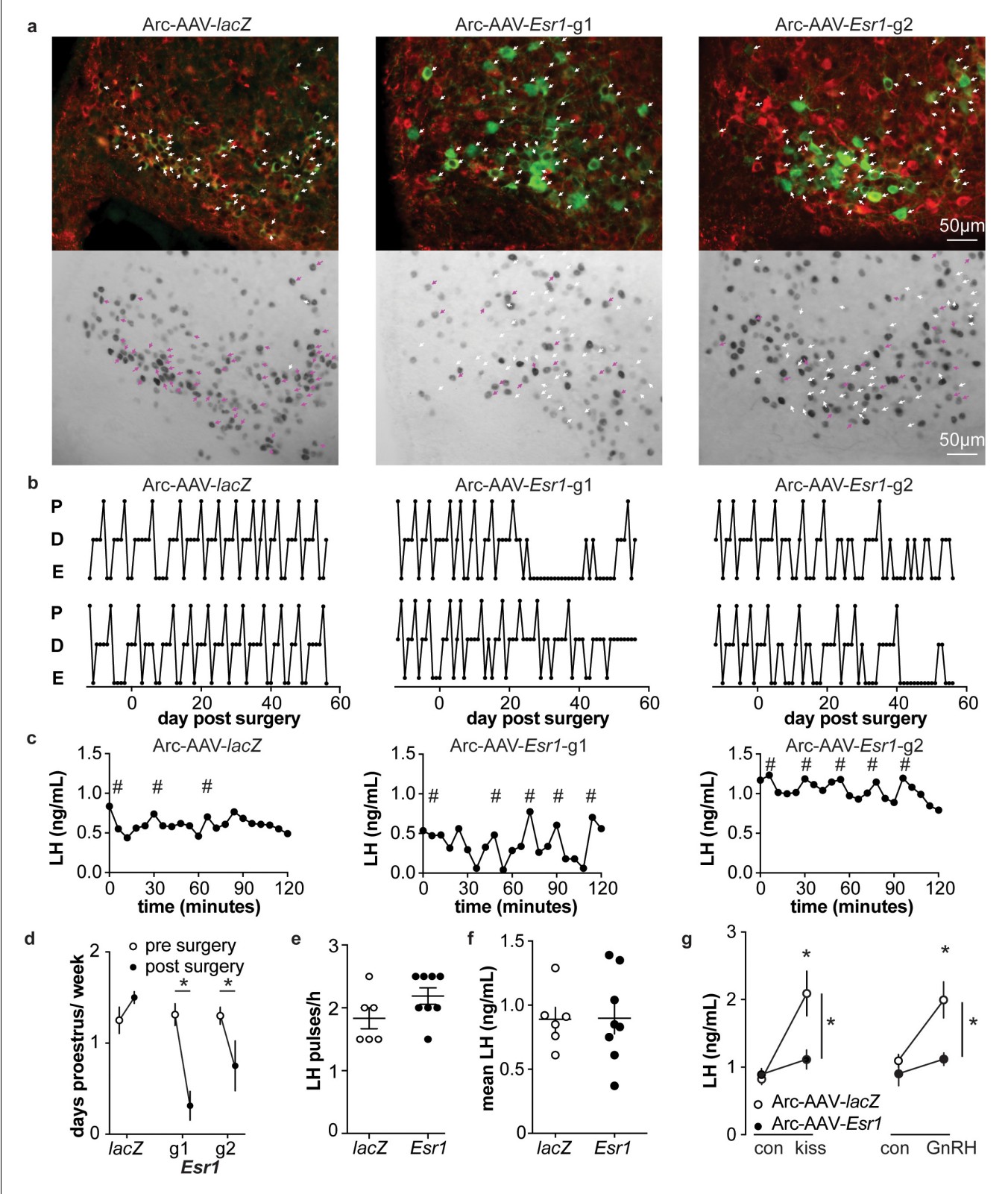

**Figure 5.** Deletion of ER in arcuate kisspeptin neurons. (**a**) Arc-AAV-*lacZ* and Arc-AAV-*Esr1* (g1 or g2) were bilaterally delivered to arcuate region (see *Figure 5—figure supplement 1*). Brain sections were processed to detect GFP (green), mCherry (red) and ERα (black). Arc-AAV-*Esr1* infected arcuate kisspeptin neurons exhibit decreased ERα expression compared to Arc-AAV-*lacZ* infected cells (bottom panel of each pair, white arrows indicate ERα-negative, magenta arrows indicate ERα-positive infected cells). (**b**) representative reproductive cycles of mice that received Arc-AAV-*lacZ*, -*Esr1* g1 or

*Figure 5 continued on next page*

*Figure 5 continued*

g2; E, estrus; D, diestrus; P, proestrus. Day 0 indicates the day of stereotaxic surgery. (c) pulsatile LH release in Arc-AAV-*lacZ*, -*Esr1* g1 or g2 mice, # indicate pulse detected by Cluster analysis (*Veldhuis and Johnson, 1986*). (d) Mean ± SEM days/week in proestrus before (from day −12 to day 14) and after infection (day 29 to day 56) in mice receiving Arc-AAV-*lacZ*, -*Esr1* g1 or g2. (e) Individual values and mean ± SEM LH pulses/h. (f) Individual means and mean ± SEM mean LH over the entire pretreatment sampling period. (g) Mean ± SEM LH before (con) and 15 min after kisspeptin (kiss) injection (left) and before (con) and 15 min after GnRH injection (right). *p<0.05.
DOI: https://doi.org/10.7554/eLife.43999.014

The following figure supplement is available for figure 5:

**Figure supplement 1.** Bilateral delivery of AAV-*lacZ*, and AAV-*Esr1* (g1 and g2) to arcuate of adult female mice.
DOI: https://doi.org/10.7554/eLife.43999.015

## Knockdown of ERα in arcuate kisspeptin neurons in adulthood increase ionotropic glutamatergic transmission to these cells but does not alter their short-term spontaneous firing rate

Arcuate kisspeptin neurons are postulated to form an interconnected network that is steroid sensitive and utilizes glutamatergic transmission at least in part for intranetwork communication (*Qiu et al., 2016*). We thus hypothesized that loss of ERα specifically from arcuate kisspeptin neurons would increase their spontaneous firing rate and increase glutamatergic transmission to these cells, similar to what is observed in these cells in KERKO mice (*Wang et al., 2018*). As Arc-AAV-*Esr1* knockdown mice spend the most time in estrus, similar to KERKO mice, we used estrus as the reproductive stage to examine the short-term (~10 min) firing frequency of these neurons and AMPA-mediated excitatory glutamatergic postsynaptic currents (EPSCs). The firing frequency of Arc-AAV-*Esr1* infected cells was not different from Arc-AAV-*lacZ* infected cells (*Figure 6a,b*, Mann-Whitney U-test, p=0.14) even though there tends to be more cells firing at >1 Hz in the Arc-AAV-*Esr1* group compared to the Arc-AAV-*lacZ* group (*Figure 6c*, Fisher's exact test, *Esr1* vs *lacZ*, p=0.07). In contrast, the frequency and amplitude of glutamatergic EPSCs in arcuate kisspeptin cells in Arc-AAV-*Esr1* infected mice was greater than in the Arc-AAV-*lacZ* group (*Figure 6d,e,f*, two-tailed unpaired Student's t-test, frequency p=0.0007, amplitude p=0.014).

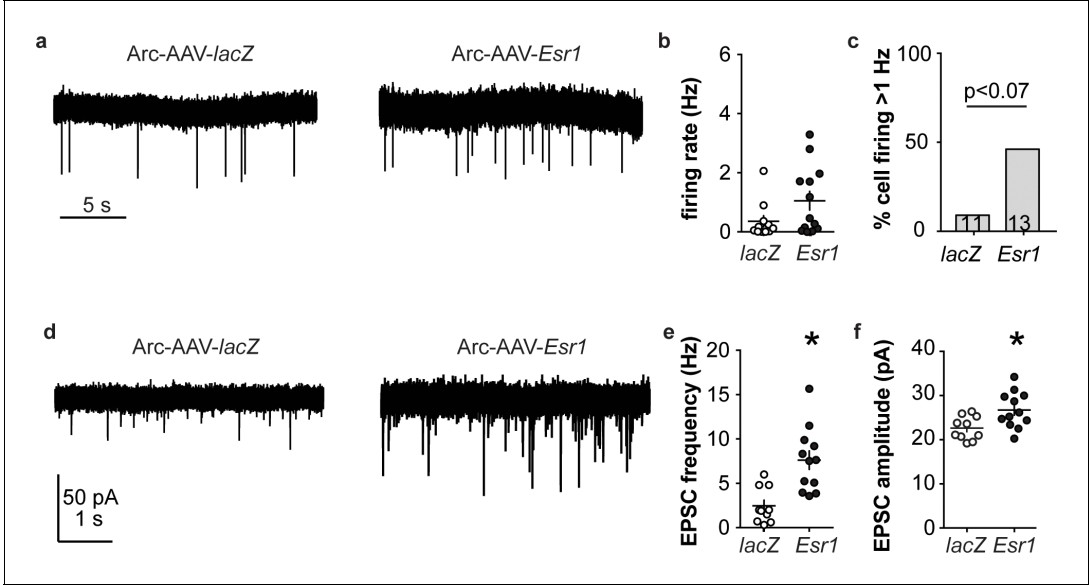

**Figure 6.** *Esr1* knockdown in arcuate kisspeptin neurons alters cellular physiology. (a) Representative extracellular recordings of firing rate. (b), (c) individual values and mean ± SEM firing rate (b) and percent of cells with firing rate >1 Hz (c); cells per group shown in bars. (d) representative whole-cell recordings of EPSCs. (e, f) individual values and mean ± SEM of EPSC frequency (e) and amplitude (f). *p<0.05.
DOI: https://doi.org/10.7554/eLife.43999.016

**Table 3.** Specificity of the *Esr1* sgRNAs and off-target predictions by Feng Zhang's guide design tool software (http://crispr.mit.edu); Benchling analysis (https://benchling.com/academic) produced a subset of these results.

| sgRNA | *Specificity score | & mismatches (MMs) between sgRNA and gene locus | Gene | | # Off-target score | Locus |
|---|---|---|---|---|---|---|
| *Esr1*-g1 | 90 | 4MMs [2:9:11:12] | NM_013870 | *Smtn* | 0.2 | chr11:+3417882 |
| | | 4MMs [5:10:13:19] | NM_009728 | *Atp10a* | 0.2 | chr7:−66040030 |
| | | 4MMs [4:9:15:20] | NM_023805 | *Slc38a3* | 0.1 | chr9:+107561207 |
| | | 4MMs [7:8:15:19] | NM_001037764 | *Rai1* | 0.1 | chr11:+60003351 |
| | | 4MMs [3:10:13:14] | NM_053193 | *Cpsf1* | 0.1 | chr15:−76426196 |
| *Esr1*-g2 | 73 | 4MMs [4:8:11:12] | NM_001024560 | *Snx32* | 0.4 | chr19:+5495979 |
| | | 4MMs [2:4:5:16] | NM_001194923 | *Cldn18* | 0.3 | chr9:+99617489 |

*Values range from 1 to 100 index to assess the specificity of a guide, with 100 being the most specific guide.

&4MMs [2:9:11:12] indicates nucleotides 2, 9, 11, 12 of the sgRNA do not match the 'off target' gene locus.

#Off-target score values range from 0 to 100, with 100 being the value for the target *Esr1* gene.

DOI: https://doi.org/10.7554/eLife.43999.010

## Discussion

This study examined the roles of two hypothalamic kisspeptin neuronal populations in mediating estradiol feedback from cellular, molecular and whole-body physiology perspectives. We utilized both conventional kisspeptin-specific ERα knockout mice (KERKO) and CRISPR-Cas9-based viral vector-mediated knockdown of *Esr1*. The latter approach allows both temporal control and nucleus-

**Table 4.** Primer probes used for single-cell qPCR.

| IDT prime time qPCR probe assay | Transcript | Forward 5'−3' | Reverse 5'−3' | Probe 5'−3' | Amplicon (bp) | Accession no. | Location |
|---|---|---|---|---|---|---|---|
| Mm.PT.58.42702897 | *Cacna1g* | CTCAACTGTA TCACCATCGCTA | AAGACTGCCG TGAAGATGT | CGCCCCAAAA TTGACCCCCAC | 101 | NM_009783 | 4446–4546 |
| Mm.PT.58.15908160 | *Cacna1h* | GACACTGTGG TTCAAGCTCT | TTATCCTCGC TGCATTCTAGC | ACCTTGGTCTTCT TTTCATGCTCCTGT | 122 | NM_021415 | 5565–5686 |
| Mm.PT.58.9567566 | *Cacna1i* | CATCACCTTC ATCATCTGCCT | CCTCCAGCA CAAAGACAGT | ACCAGCCTACAT CCCTAGAGACAGC | 125 | NM_001044308 | 4914–5038 |
| Mm.PT.58.41764708 | *Esr1* | GCTCCTTCTC ATTCTTTCCCA | TCCAGGAGC AGGTCATAGAG | CCATGCCTTTG TTACTCATGTGCCG | 108 | NM_007956 | 1768–1865 |
| Mm.PT.58.16981577 | *Esr2* | CCTCCTGATGC TTCTTTCTCAT | TCGAAGC GTGTGAGCATTC | TCCATGCCCTTG TTACTGATGTGCC | 133 | NM_207707 | 1829–1961 |
| Mm.PT.58.30501833 | *Hcn1* | GCGTTATCACC AAGTCCAGTA | CAGTAGGTA TCAGCTCGGACA | CTCCGAAGTAAG AGCCATCTGTCAGC | 115 | NM_010408 | 1913–2027 |
| Mm.PT.58.7963736 | *Hcn2* | CTTCACCAAG ATCCTCAGTCTG | GGTCGTAGG TCATGTGGAAA | TGCGGCTATCA CGGCTCATCC | 98 | NM_008226 | 935–1032 |
| Mm.PT.58.7999585 | *Hcn3* | GCCTCACTGA TGGATCCTACT | TCAAGCACC GCATTGAAGT | ACCTATTGTCG CCTCTACTCGCTCA | 130 | NM_008227 | 1546–1675 |
| Mm.PT.58.43863085 | *Hcn4* | GCTGATGGC TCCTATTTTGGA | TCATTGAA GTTGTCCACGCT | AAGTATCCGC TCTGACGCTGGC | 116 | NM_001081192 | 2614–2729 |
| Mm.PT.45.16269514 | *Kiss1* | CTGCTTCTCC TCTGTGTCG | TTCCCAGG CATTAACGAGTTC. | CGGACTGCTG GCCTGTGGAT | 105 | NM_178260 | 66–170 |
| Mm.PT.47.10254276 | *Pgr* | CGCCATACCTT AACTACCTGAG | CCATAGTGA CAGCCAGATGC | AGATTCAGAAGC CAGCCAGAGCC | 124 | NM_008829 | 2230–2353 |
| Mm.PT.51.17048009.g | *Syn1* | CTTGAGCAGA TT GCCATGTC | ACCTCAATAAT GTGATCCCTTCC | ACGTGTCTACCC ACAACTTGTACCTG | 131 | NM_013680 | 1159–1289 |
| Mm.PT.58.33106186 | *Th* | CCCTACCAAGA TCAAACCTACC | CTGGATACGAG AGGCATAGTTC | TGAAGCTCTCTG ACACGAAGTACACCG | 96 | NM_009377 | 1298–1393 |

DOI: https://doi.org/10.7554/eLife.43999.013

specific manipulations to distinguish the role of ERα within each population in negative and positive feedback regulation of LH release and neurobiological properties (*Figure 7*).

AVPV kisspeptin neurons are postulated to convey estradiol positive feedback signals to generate the GnRH surge. Consistent with this postulate, these neurons are more excitable during positive feedback and also receive increased glutamatergic transmission (*Piet et al., 2013*; *Zhang et al., 2013*; *Wang et al., 2016*; *Wang et al., 2018*). AVPV kisspeptin cells in both KERKO and AVPV-AAV-*Esr1* models are less excitable compared to controls, firing fewer bursts and single action potentials in response to the same current injection. Results from the AVPV-AAV-*Esr1* model support and extend data from KERKO mice and provide evidence towards accepting the hypothesis that the role of ERα in shifting excitability is activational, independent of its role in the development of these cells (*Mayer et al., 2010*).

Kisspeptin expression in AVPV cells is estradiol activated and fewer cells expressing *Kiss1* mRNA are detected in this region in KERKO mice (*Greenwald-Yarnell et al., 2016*). In AVPV-AAV-*Esr1* mice with adult knockdown, we also observed fewer cells express *Kiss1* mRNA compared to AVPV-AAV-*lacZ* mice, further supporting an activational role for estradiol in the adult physiology of these cells. The inability to sense estradiol through ERα in AVPV kisspeptin neurons may reduce production and release of kisspeptin and ultimately impair the downstream GnRH/LH surge. This could explain the blunted LH surges in AVPV-AAV-*Esr1* injected mice. *Esr1* knockdown in the AVPV, however, did not alter reproductive cyclicity monitored by changes in vaginal cytology. Vaginal cytology

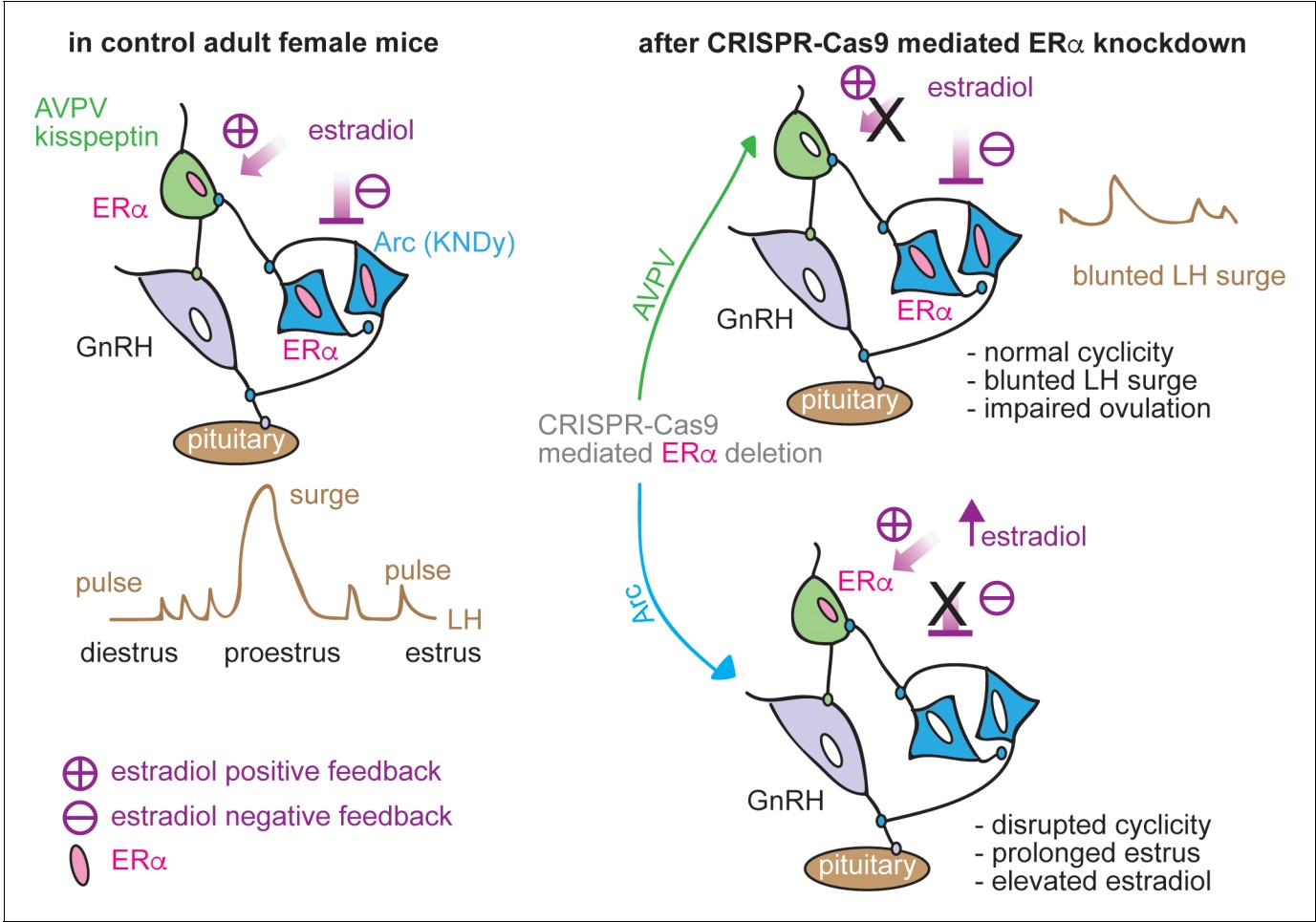

**Figure 7.** Schematic diagram of estradiol feedback regulation on ERα in AVPV and arcuate kisspeptin neurons in adulthood. Knockdown of ERα in AVPV kisspeptin neurons blunted LH surge but did not alter reproductive cyclicity whereas knockdown of ERα in arcuate kisspeptin neurons disrupted the cyclicity.

DOI: https://doi.org/10.7554/eLife.43999.017

reflects circulating steroids, in particular estradiol. Of note, only some of these mice had evidence of typical ovulation monitored by the number of corpra lutea. It is possible that sufficient estradiol is produced during these cycles to induce vaginal cytology, but not to trigger an LH surge. It is important to point out, however, that estradiol-induced LH surges, in which an established dose of estradiol was provided to the mouse, were also blunted in AVPV-AAV-*Esr1* mice. This latter observation suggests that inappropriate response of the neuroendocrine system to estradiol is, at least in part, responsible for the blunting of the LH surge. With regard to the continuation of estrous cycles in the AVPV-AAV-*Esr1* mice, typical function of the remaining ERα-positive AVPV kisspeptin neurons may be sufficient to drive maintain cyclicity. Alternatively, cyclicity and the associated changes in sex steroids may be controlled by other cells that express ERα. Of interest, stress or neuronal androgen receptor KO can similarly disrupt the LH surge without a change in estrous cyclicity (*Wagenmaker and Moenter, 2017*; *Walters et al., 2018*).

In support of a non-AVPV kisspeptin neuronal population being a primary driver of estrous cyclicity, several reproductive phenotypes of KERKO and AVPV-targeted ERα knockdown mice are different. KERKO mice tend to exhibit prolonged vaginal cornification and enlarged uteri, neither of which were observed in mice in which AAV-*Esr1* infection was targeted to the AVPV (*Figure 1—figure supplement 1c*). In contrast, prolonged estrus and enlarged uteri were observed in mice in which Arc-AAV-*Esr1* infection was targeted to arcuate kisspeptin neurons. These latter neurons have been postulated to play a role in generating episodic GnRH output. Changes in episodic GnRH frequency drive gonadotropins and thus follicle development and steroidogenesis, including the estradiol rise, which triggers positive feedback and changes in vaginal cytology. Long-term firing output of arcuate kisspeptin neurons in brain slices is episodic and steroid modulated (*Vanacker et al., 2017*), and activation of these cells in vivo generates a pulse of LH release (*Clarkson et al., 2017*). Further evidence comes from *Tac2*-specific ERα KO mice, in which ERα is primarily deleted from the arcuate, not the AVPV, kisspeptin population. These mice also exhibit prolonged vaginal cornification (*Greenwald-Yarnell et al., 2016*). We thus hypothesize that ERα in arcuate kisspeptin neurons contributes to maintaining pulsatile LH release and mediates central estradiol negative feedback.

Consistent with this postulate, partial (~65%) adult knockdown of ERα in these cells altered the reproductive cycle. As KERKO mice exhibit increased LH-pulse frequency, we were initially surprised we did not observe differences in pulse frequency or mean LH levels in mice receiving Arc-AAV-*Esr1*. This may be attributable to single housing conditions in the present experiment, which may make mice prone to stress despite more than four weeks of handling before sampling. It is also possible that the pulse frequency during diestrus differs between Arc-AAV-*Esr1* and Arc-AAV-*lacZ* mice. Despite this lack of statistical difference in LH-pulse frequency, ERα knockdown mice had a markedly reduced response to IP injection of both kisspeptin and GnRH, similar to KERKO mice (*Wang et al., 2018*). This suggests loss of ERα function in arcuate kisspeptin neurons may disrupt GnRH neuronal response to kisspeptin and/or the pituitary response to GnRH. This could arise from a disruption of negative feedback leading to overstimulation and thus desensitization of the hypothalamo-pituitary-gonadal axis or blunting of the response to administered neuropeptides.

Dissection of the electrophysiological properties of arcuate kisspeptin neurons revealed that glutamatergic transmission to these neurons was elevated when ERα is knocked down. This indicates the connectivity of these cells remains plastic even after puberty. The observation that targeted reduction of ERα in arcuate kisspeptin neurons increases glutamatergic transmission further suggests interconnections among these cells provide many of their glutamatergic inputs. Given this, it is intriguing that the short-term firing rate of these cells was not increased, although there was a strong trend toward a greater percent of higher frequency cells. The lack of change in mean firing rate may reflect the partial deletion of ERα in this population, with lower firing rate being preserved in cells with ERα, and elevated EPSC frequency arising at least in part from the high firing cells. It is also possible that long-term firing patterns of these Arc-AAV-*Esr1* infected arcuate kisspeptin neurons, which may be associated with episodic neuroendocrine activity, are disrupted. These data support the idea that glutamatergic inputs to arcuate kisspeptin neurons play an important role on maintaining normal reproductive function.

Although the CRISPR-Cas9-based knockdown approach allows spatial and temporal control, it too has caveats. For example, sgRNAs may have off-target actions on other regions of the genome beyond the sites predicted by the design software (*Anderson et al., 2018*). To address this, we independently tested two sgRNAs that target *Esr1* to address the possible off-target effects among

groups. We did not observe any differences between *Esr1* guide1 and guide2 groups. This suggests the phenotypes observed are primarily attributable to the deletion of ERα. Because of the nature of the nonhomologous end joining repair machinery activated after CRISPR-Cas9-initiated cuts, *Esr1* gene editing in each cell varies. It is difficult to assess each individual neuron to test if mutations at other genes are potentially involved in changes of biophysical properties. Despite these variables, in the present study the systemic and cellular phenotypes in *Esr1* guide1 vs guide2 infected mice were quite consistent.

In conclusion, utilizing CRISPR-Cas9 AAV, we were able to successfully knockdown ERα in specific populations of kisspeptin neurons in adult female mice. Knockdown in each population recapitulated part of the KERKO model and furthers our understanding the role ERα in that population in regulating estradiol feedback.

# Materials and methods

**Key resources table**

| Reagent type (species) or resource | Designation | Source or reference | Identifiers | Additional information |
|---|---|---|---|---|
| Mus musculus, C57BL/6J | Kiss1-ires-Cre | PMID 26862996 | | Dr. Martin Myers, University of Michigan |
| Mus musculus, C57BL/6J | Esr1 loxp | PMID 17785410 | | Dr. Martin Myers, University of Michigan |
| Mus musculus, C57BL/6J | Kiss1-cre | Dr. Carol Elias/Jackson Labs | JAX 023426 | |
| Mus musculus, C57BL/6J | Cas9-stop loxp | Jackson Labs | Jax 024858; RRID:IMSR_JAX:024858 | |
| Mus musculus, C57BL/6J | Rosa26-EYFP | Jackson Labs | Jax 006148; RRID:IMSR_JAX:006148 | |
| Mus musculus myoblast, C3H | C2C12 myoblast | ATTC | Cat # CRL 1772 | Dr. Daniel Michele, University of Michigan |
| Antibody | rabbit anti-ERα | Millipore | #06-935 | dil. 1:10000 |
| Antibody | rat anti-mCherry | Invitrogen | M11217 | dil. 1:5000 |
| Antibody | chicken anti-GFP | Abcam | ab13970 | dil. 1:2000 |
| Recombinant DNA reagent | AAV8-hsyn-dio-sgRNA_lacZ-mCherry | this paper | Custom Order | UNC-viral core |
| Recombinant DNA reagent | AAV8-hsyn-dio-sgRNAEsr1_g1-mCherry | this paper | Custom Order | UNC-viral core |
| Recombinant DNA reagent | AAV8-hsyn-dio-sgRNAEsr1_g2-mCherry | this paper | Custom Order | UNC-viral core |
| Recombinant DNA reagent | plasmid LentiV2-sgRNA-Esr1_g1 | this paper | built on lentiCRISPRv2; Addgene Cat #52961 | |
| Recombinant DNA reagent | plasmid LentiV2-sgRNA-Esr1_g2 | this paper | built on lentiCRISPRv2; Addgene Cat #52961 | |
| Recombinant DNA reagent | plasmid LentiV2-sgRNA-lacZ | this paper | built on lentiCRISPRv2; Addgene Cat #52961 | |
| Commercial assay or kit | ABC amplification | Vector Laboratories | Cat # PK-6100 | |
| Chemical compound, drug | CNQX | Sigma-Aldrich | Cat # 1045 | |
| Chemical compound, drug | APV | Tocris | Cat # 0106 | |
| Chemical compound, drug | picrotoxin | Sigma-Aldrich | Cat # P1675 | |
| Chemical compound, drug | TTX | Tocris | Cat # 1069 | |

*Continued on next page*

*Continued*

| Reagent type (species) or resource | Designation | Source or reference | Identifiers | Additional information |
|---|---|---|---|---|
| Chemical compound, drug | 10% Neutral Buffered Formalin | Fisher Scientific | Cat # 22899402 | |
| Chemical compound, drug | Hydrogen Peroxide | Sigma | Cat # 216763 | |
| Chemical compound, drug | LHRH | Bachem | Cat # H4005 | |
| Chemical compound, drug | kisspeptin | Phoenix | Cat # 048-56 | |
| Chemical compound, drug | Neurobiotin | Vector Labs | Cat # SP-1120 | |
| Software, algorithm | Igor Pro | Wavemetrics | https://github.com/defazio2/LWeLifeRepo | |

## Animals

The University of Michigan Institutional Animal Care and Use Committee approved all procedures. Adult female mice (60–150 days) were used. Mice were provided with water and Harlan 2916 chow (VetOne) *ad libitum* and were held on a 14L:10D light cycle (lights on 0400 Eastern Standard Time). To delete ERα specifically from all kisspeptin cells (*Wang et al., 2018*), mice with the *Cre* recombinase gene knocked-in after the *Kiss1* promoter (*Kiss1*-ires-Cre mice, *Cravo et al., 2011*) were crossed with mice with a floxed *Esr1* gene, which encodes ERα (ERα floxed mice) (*Greenwald-Yarnell et al., 2016*). The expression of Cre recombinase mediates deletion of ERα in kisspeptin cells (KERKO mice). To visualize kisspeptin neurons for recording, mice heterozygous for both Kiss-Cre and floxed ERα were crossed with Cre-inducible YFP mice. Crossing mice heterozygous for all three alleles yielded litters that contained some mice that were homozygous for floxed ERα and at least heterozygous for both *Kiss1*-Cre and YFP; these were used as KERKO mice. Littermates of KERKO mice with wild-type *Esr1*, *Kiss1*-Cre YFP (heterozygous or homozygous for either Cre or YFP) were used as controls; no differences were observed among these controls and they were combined.

To generate kisspeptin-specific *S. pyogenes* Cas9 (Cas9)-expressing mice, mice with the *Cre* recombinase gene knocked-in after the *Kiss1* promoter (*Kiss1-Cre* mice) were crossed with mice that have Cre recombinase-dependent expression of CRISPR-associated protein 9 (Cas9) endonuclease, a 3X-FLAG epitope tag and eGFP directed by a CAG promoter. KERKO mice have disrupted estrous cycles with persistently cornified vaginal cytology typical of estrus; we thus used females in estrus as controls. Estrous cycle stage was determined by vaginal lavage. To examine the role of circulating estradiol, mice were ovariectomized (OVX) under isoflurane anesthesia (Abbott) and were either simultaneously implanted with a Silastic (Dow-Corning) capsule containing 0.625 μg of estradiol suspended in sesame oil (OVX +E) or not treated further (OVX) (*Christian et al., 2005*). Bupivacaine (0.25%, APP Pharmaceuticals) was provided local to incisions as an analgesic. These mice were studied 2-3 days after surgery. Mice for electrophysiology were sacrificed at the time of estradiol positive feedback in the late afternoon (*Christian et al., 2005*). For free-floating immunochemistry staining, mice were perfused at 1700 EST 2-3d post OVX +E surgery at the expected peak of the estradiol-induced LH surge.

## sgRNA design

For Cas9 target selection and generating single guide RNAs (sgRNA), 20-nt target sequences were selected to precede a 5′NGG protospacer-adjacent motif (PAM) sequence. To minimize off-targeting effects and maximize sgRNA activity, two CRISPR design tools were used to evaluate sgRNAs (*Ran et al., 2013*; *Doench et al., 2014*) targeting the first coding exon of mouse *Esr1*. The two best candidates were selected based on lowest predicted off-target effects and highest activity. The target sequence for guide 1 is 5′-CACTGTGTTCAACTACCCCG-3′ (referred to as g1) and the target sequence for guide 2 is 3′-CTCGGGGTAGTTGAACACAG-5′ (referred to as g2). Because g1 and g2

were similarly effective in *Esr1* knockdown and effects on cycles, mice were combined for physiology studies. Control sgRNA sequence was designed to target *lacZ* gene from *Escherichia coli* (target sequence: 5'-TGCGCAGCCTGAATGGCGAA −3').

## In vitro validation of sgRNAs

Mycoplasma-free C2C12 mouse myoblast cells (generous gift of Dr. Daniel Michele, University of Michigan) were grown in DMEM containing 10% FBS (Thermo Fisher) at 37°C in 5% CO2. Each individual sgRNA was introduced to BsmBI site of the lentiCRISPRv2 construct. Cells were co-transfected with one of the lentiCRISPRv2 plasmids containing sgRNAs and a standard GFP plasmid construct (*Ramakrishnan et al., 2016*) using Lipofectamine 3000 (Invitrogen) according to the manufacturer's instructions. Cells were selected for ~4 weeks with medium containing 1 μg/mL puromycin. Selected cells were harvested, DNA isolated using the Qiagen DNA Extraction Kit, and sequenced with primers for *Esr1*.

## AAV vector production

To construct the AAV plasmid, a mCherry-U6 promoter-sgRNA scaffold segment was synthesized by Integrated DNA Technologies (IDT). After PCR amplification, the ligation product containing mCherry-U6 promoter-sgRNA scaffold was cloned in reverse orientation into a hSyn (human Synapsin 1) promoter driven *Cre*-inducible AAV vector backbone (*Flak et al., 2017*). The individual sgRNAs (with an extra G added to the 5'-end of each sgRNA to increase guide efficiency [*Doench et al., 2014*]) were then inserted into a designed SapI site between U6 promoter and sgRNA scaffold component. All three AAV viral vectors were prepared in AAV8 serotype at University of North Carolina Vector Core.

## Stereotaxic injections

Kiss1Cre/Cas9-GFP female animals (>2 mo) were checked for estrous cycles for >10 days before surgery; only mice with regular 4–5 day cycles were used. Mice were anesthetized with 1.5–2% isoflurane. AVPV injections were targeted to 0.55 mm posterior to Bregma, ±0.2 mm lateral to midline, and 4.7 and 4.8 mm ventral to dura. Arcuate injections were targeted to 1.5–1.7 mm posterior to Bregma, ±0.2 mm lateral to midline, and 5.9 mm ventral to dura. 100 nl virus injected bilaterally at the target coordinates at ~5 nl/min. The pipette was left in place for 5 min after injection to allow viral diffusion into the brain. Carprofen (Zoetis, Inc., 5 mg/kg, sc) was given before and 24 hr after surgery to alleviate postsurgical pain. Estrous cycle monitoring continued after surgery for up to 8 weeks. Stereotaxic hits were defined as ≥70% infection rate in both hemispheres; only bilateral hits were included for in vivo evaluation of reproductive parameters.

## Perfusion and free-floating immunohistochemistry

Mice were anesthetized with isoflurane and then transcardially perfused with PBS (15–20 mL) then 10% neutral-buffered formalin for 10 min (~50 mL). Brains were placed into the same fixative overnight, followed by 30% sucrose for ≥24 hr for cryoprotection. Sections (30 μm, four series) were cut on a cryostat (Leica CM3050S) and stored at −20°C in antifreeze solution (25% ethylene glycol, 25% glycerol in PBS). Sections were washed with PBS, treated with 0.1% hydrogen peroxide, and then placed in blocking solution (PBS containing 0.1% TritonX-100, 4% normal goat serum, Jackson Immunoresearch) for 1 hr at room temperature, then incubated with rabbit anti-ERα (#06–935, Millipore, 1:10,000; this antibody recognizes the C-terminus of ERα.) in blocking solution 48 hr at 4°C. Sections were washed then incubated with biotinylated anti-rabbit antibody (Jackson Immunoresearch, 1:500) followed by ABC amplification (Vector Laboratories, 1:500) and nickel-enhanced diaminobenzidine (Thermo Scientific) reaction (4.5 min). Sections were washed with PBS and incubated overnight with chicken anti-GFP (ab13970, Abcam, 1:2000) and rat anti-mCherry (M11217, Invitrogen, 1:5000) in blocking solution. The next day, sections were washed and incubated with Alexa 594-conjugated anti-rat and Alexa 488-conjugated anti-chicken antibodies for 1 hr at room temperature (Molecular Probes, 1:500). Sections were mounted and coverslipped (VWR International 48393 251). Images were collected on a Zeiss AXIO Imager M2 microscope, and the number of immunoreactive GFP only, GFP/mCherry, and GFP/mCherry/ERα cells were counted in

the injected region. The other kisspeptin region in the hypothalamus was examined and no infection of kisspeptin cell bodies was observed.

## Brain slice preparation

All solutions were bubbled with 95%$O_2$ and 5%$CO_2$ for $\geq$15 min before exposure to tissue and throughout experiments. Brains were rapidly removed 1.5–2 hr before lights off and placed in ice-cold sucrose saline solution containing (in mM): 250 sucrose, 3.5 KCl, 25 NaHCO$_3$, 10 D-glucose, 1.25 Na$_2$HPO$_4$, 1.2 MgSO$_4$, and 3.8 MgCl$_2$. Coronal slides (300 μm) were made with a Leica VT1200S. Slices were incubated in a 1:1 mixture of sucrose-saline and artificial cerebrospinal fluid (ACSF) containing (in mM): 135 NaCl, 3.5 KCl, 26 NaHCO$_3$, 10 D-glucose, 1.25 Na$_2$HPO$_4$, 1.2 MgSO$_4$, 2.5 CaCl$_2$ for 30 min at room temperature. Slices were then transferred to 100% ACSF at room temperature for $\geq$30 min before recording. Slices were used within 6 hr of preparation.

## Electrophysiology recordings

Slices were transferred to a recording chamber and perfused with oxygenated ACSF (3 mL/min) and heated by an in-line heater (Warner Instruments) to 30 ± 1°C. GnRH-GFP neurons were identified by brief illumination at 470 nm using an upright fluorescence microscope Olympus BX51W1. Recording pipettes were pulled from borosilicate glass (type 7052, 1.65 mm outer diameter and 1.12 mm inner diameter; World Precision Instruments, Inc) using a P-97 puller (Sutter Instruments) to obtain pipettes with a resistance of 2–3.5 MΩ. Recordings were performed with an EPC-10 dual-patch clamp amplifier and Patchmaster acquisition software (HEKA Elektronik). Recorded cells were mapped to a brain atlas (*Paxinos and Franklin, 2001*) to determine if cell location was related to response to treatment. No such correlation was observed in this study.

## Extracellular recordings

Extracellular recordings were used to characterize firing rate as they maintain internal milieu and have minimal impact neuronal firing rate (*Nunemaker et al., 2003*; *Alcami et al., 2012*). Recordings were made with receptors for ionotropic GABA$_A$ and glutamate synaptic transmission antagonized (100 μM picrotoxin, 20 μM D-APV [D-(−)−2-amino-5-phosphonopentenoic acid], 10 μM CNQX [6-cyano-7-nitroquinoxaline]). Pipettes were filled with HEPES-buffered solution containing (in mM): 150 NaCl, 10 HEPES, 10 D-glucose, 2.5 CaCl$_2$, 1.3 MgCl2, and 3.5 KCl (pH = 7.4, 310 mOsm), and low-resistance (22 ± 3 MΩ) seals formed between the pipette and neuron after first exposing the pipette to the slice tissue in the absence of positive pressure. Recordings were made in voltage-clamp mode (0 mV pipette holding potential) and signals acquired at 20 kHz and filtered at 10 kHz. Resistance of the loose seal was checked frequently during the first 3 min of recordings to ensure a stable baseline, and also before and after a subsequent 10 min recording period; data were not used if seal resistance changed >30% or was >25 MΩ. The first 5 min of this 10-min recording were consistently stable among cells and were thus used for analysis.

## Whole-cell recordings

For whole-cell patch-clamp recordings, three different pipette solutions were used depending on the goal. Most recordings were done with a physiologic pipette solution containing (in mM): 135 K gluconate, 10 KCl, 10 HEPES, 5 EGTA, 0.1 CaCl$_2$, 4 MgATP and 0.4 NaGTP, pH 7.2 with NaOH, 302 ± 3 mOsm. A similar solution containing 10 mM neurobiotin was adjusted to similar osmolarity. A solution in which cesium gluconate replaced potassium gluconate was used to reduce potassium currents and allow better isolation of calcium currents. Membrane potentials reported were corrected online for liquid junction potential of −15.7 mV, same among all solutions (*Barry, 1994*).

After achieving a minimum 1.6 GΩ seal and the whole-cell configuration, membrane potential was held at −70 mV between protocols during voltage-clamp recordings. Series resistance (R$_s$), input resistance (R$_{in}$), holding current (I$_{hold}$) and membrane capacitance (C$_m$) were frequently measured using a 5 mV hyperpolarizing step from −70 mV (mean of 16 repeats). Only recordings with R$_{in}$ >500 MΩ, I$_{hold}$−40 to 10 pA and R$_S$ <20 MΩ, and stable C$_m$ were accepted. R$_s$ was further evaluated for stability and any voltage-clamp recordings with ΔR$_s$ >15% were excluded; current-clamp recordings with ΔR$_s$ >20% were excluded. There was no difference in I$_{hold}$, C$_m$, or R$_s$ among any comparisons.

For current-clamp recordings, depolarizing and hyperpolarizing current injections (−50 to +50 pA, 500 ms, 10 pA increments) were applied from an initial membrane potential of −71 ± 2 mV, near the resting membrane potential of these cells (*DeFazio et al., 2014*).

For voltage-clamp recordings of excitatory postsynaptic currents (EPSCs), membrane potential was held at −68 mV, the reversal potential for GABA$_A$-receptor mediated currents, and ACSF contained picrotoxin (100 μM), and D-APV (20 μM).

For voltage-clamp recordings of I$_T$, ACSF containing antagonists of ionotropic GABA$_A$ and glutamate receptors was supplemented with TTX (2 μM) and the Cs-based pipette solution was used. Two voltage protocols were used to isolate I$_T$ as reported (*Wang et al., 2016*). First, total calcium current activation was examined. Inactivation was removed by hyperpolarizing the membrane potential to −110 mV for 350 ms (not shown in figures). Next, a 250 ms prepulse of −110 mV was given. Then membrane potential was varied in 10 mV increments for 250 ms from −110 to −30 mV. Finally, test pulse of −40 mV for 250 ms was given. From examination of the current during the test pulse, it was evident that no sustained (high-voltage activated, HVA) calcium current was activated at potentials more hyperpolarized than −40 mV. To remove HVA contamination from the step to −30 mV, a second protocol was used in which removal of inactivation (−110 mV, 350 ms) was followed by a 250 ms prepulse at −40 mV, then a step for 250 ms at −30 mV and finally a test pulse of −40 mV for 250 ms. I$_T$ was isolated by subtracting the trace following the −40 mV prepulse from those obtained after the −110 mV prepulse for the depolarized variable step to −30 mV; raw traces from the initial voltage protocol were used without subtraction for variable steps from −110 mV to −40 mV because of the lack of observed activation of HVA at these potentials. Activation of I$_T$ was assessed from the resulting family of traces by peak current during the variable step phase. Inactivation of I$_T$ was assessed from the peak current during the final −40 mV test pulse.

## *Post hoc* identification of ERα

The pipette solution containing neurobiotin was used for recordings cells from AAV-injected mice. An outside-out patch was formed after recording to reseal the membrane and the location of cells was marked on a brain atlas (*Paxinos and Franklin, 2001*). The brain slices were fixed overnight in 10% formalin at 4°C and changed to PBS. Slices were photo-bleached with a UV illuminator for ~72 hr and checked to ensure no visible fluorescent signal was observed. Slices were then placed in blocking solution for 1 hr, then incubated with rabbit anti-ERα for 48 hr at 4°C. Slices were washed and then incubated with Alexa 594-conjugated anti-rabbit and Alexa 350-conjugated neutravidin for 2 hr at room temperature (Molecular Probes, 1:500). Slices were mounted, coverslipped and imaged as above. Cells with neurobiotin-labeling were examined for ERα-immunoreactivity.

*Single-cell PCR* Cells for single cell PCR were collected as previously described (*Ruka et al., 2013*). Patch pipettes (2–3 MΩ) were filled with 5–8 μL of an RNase free solution containing (in mM): 135 K-gluconate, 10 KCl, 10 HEPES, 5 EGTA, 4.0 Mg-ATP, 0.4 Na-GTP, and 1.0 CaCl$_2$ (pH 7.3, 305 mOsm). Additionally, just before use 1 U/μL Protector RNase Inhibitor (Roche, Indianapolis, IN) was added to the pipette solution. Single-cell RNA was harvested from the target cells in whole-cell configuration after recording membrane response in current-clamp; cytoplasm was aspirated into the pipette and expelled into a 0.2 mL tube containing reverse transcriptase buffer (Superscript Vilo cDNA Synthesis Kit, Invitrogen/ThermoFisher), volume was adjusted to 20 μL with molecular grade water. Cell contents were reverse transcribed following manufacturer's instructions. False harvests, in which the pipette was lowered into the slice preparation but no aspiration of cell contents occurred, were used to estimate background contamination. These were performed on each recording day. Additionally, a standard curve of mouse hypothalamic RNA (1, 0.1, 0.01, 0.001 ng/μL final concentration) and a water blank (negative control) were reverse transcribed. An equivalent volume of water or patch solution was reverse transcribed as a negative control. Single-cell cDNA, controls, and the standard curve were preamplified for 15 cycles using TaqMan PreAmp Master Mix (Invitrogen/ThermoFisher) as previously described (*Glanowska et al., 2014*). Quantitative PCR was performed using 5 μL of diluted preamplified DNA (1:10) per reaction, in duplicate, for 50 cycles (TaqMan Gene Expression Master Mix; Invitrogen). Single-cell cDNA was assayed for: *Kiss1, TH, Esr1, Esr2, Pgr, Cacna1g, Cacna1h, Cacna1i, Hcn1 Hcn2 Hcn3 Hcn4; Syn1* was used as housekeeping gene; only Syn1-positive cells were analyzed. Single cells were considered positive for a transcript if their threshold was a minimum of three cycles earlier (eight fold greater) than the false harvests and

the reverse transcribed and preamplified water blank sample. TaqMan PrimeTime qPCR assays for mRNAs (*Table 4*) were purchased from IDT.

## Tail-tip blood collection for LH pulses

Ovary-intact Kiss1Cre-Cas9 adult female mice with AAV-*lacZ* and AAV-*Esr1* targeted to the arcuate nucleus were singly-housed were handled daily ≥4 wks before sampling. Vaginal cytology was determined for ≥10 days before sampling. As the majority of AAV-*Esr1* arcuate targeted mice (6 of 9) exhibit prolonged cornification typical of estrus, all mice (*Esr1* and *lacZ*) were sampled during estrus. Repetitive tail-tip blood collecting was performed as described (*Steyn et al., 2013*). After the excision of the very tip of the tail, blood (6 µL) was collected every 6 min for 2 hr from 1pm to 3pm. At the end of this frequent sampling period, mice received a single intraperitoneal injection of kisspeptin (65 µg/kg) (*Hanchate et al., 2012*). Blood was collected just before and 15 min after kisspeptin injection. GnRH (150 µg/kg) (*Glanowska et al., 2014*) was injected 40–45 min after kisspeptin, with blood collected immediately before and 15 min after GnRH injection.

## Tail-tip blood collection for LH surge

Ovary-intact Kiss1Cre-Cas9 adult female mice with AAV-*lacZ* and AAV-*Esr1* targeted to the AVPV were singly-housed. Tail blood was collected as above on proestrus at 3, 4 and 5pm EST (lights are off at 5pm EST in the mouse room). One to two weeks later, these same mice were then subjected to OVX + E surgery and tail blood (6 µL) was collected 2–3 days post-surgery at 9am and 5pm EST.

## LH assay

Whole blood was immediately diluted in 54 µL of 0.1M PBS with 0.05% Tween 20% and 0.2% BSA, mixed and kept on ice. Samples were stored at −20˚C for a subsequent ultrasensitive LH assay (*Steyn et al., 2013*). Intraassay CV was 2.2%; interassay CVs were 7.3% (low QC, 0.13 ng/mL), 5.0% (medium QC, 0.8 ng/mL) and 6.5% (high QC, 2.3 ng/mL). Functional sensitivity was 0.016 ng/mL.

## Ovarian histology

Ovaries were fixed for 24 hr in 10% neutral-buffered formalin, then stored in 70% ethanol until paraffin embedding, sectioning (5 µm) and H and E staining. Every fifth section was examined and *corpra lutea* counted.

## Data analysis and statistics

Data were analyzed offline using custom software written in IgorPro 6.31 (Wavemetrics). For targeted extracellular recordings, mean firing rate in Hz was determined over 5 min of stable recording. In experiments examining $I_T$, the peak current amplitude at each step potential (V) was first converted to conductance using the calculated reversal potential of $Ca^{2+}$ ($E_{Ca}$) and $G = I/(E_{Ca} - V)$, because driving force was linear over the range of voltages examined. The voltage dependencies of activation and steady-state inactivation were described with a single Boltzmann distribution: $G(V) = G_{max}/(1- exp [(V_{1/2} - V_t)/k])$, where $G_{max}$ is the maximal conductance, $V_{1/2}$ is the half-maximal voltage, and k is the slope. Current density of $I_T$ at each tested membrane potential was determined by dividing peak current by membrane capacitance. LH pulses were detected by a version of Cluster (*Veldhuis and Johnson, 1986*) transferred to IgorPro using cluster sizes of two points for both peak and nadir and t-scores of two for detection of increases and decreases. Data were analyzed using Prism 7 (GraphPad Software) and reported as mean ± SEM. The number of cells per group is indicated by n and the number of mice by N in *Table 5*. For two-by-two designs, data were normally distributed and analyzed by two-way ANOVA or two-way repeated-measures (RM) with Holm-Sidak post hoc. For two group comparisons, normally-distributed data were analyzed by two-tailed unpaired Student's *t*-test; non-normal data were analyzed by Mann-Whitney U test. For categorical data, for more than three categories, *Chi*-square test of independence was used with Fisher's exact test as post hoc analysis. For two categories, Fisher's exact test was used. For each electrophysiological parameter comparison, no more than three cells per mouse was used in control and KERKO mice; no more than four cells per mouse was used for AAV-infected mice. No less than five mice were tested per parameter. The variance of the data was no smaller within an animal than among

**Table 5.** Number of cells (n) and number of mice (N) in each experiment.
For AAV-injected mice, only animals with bilateral hits are included.

| | Control | | KERKO |
|---|---|---|---|
| | **Intact n = 12, N = 7** | | **Intact n = 11, N = 6** |
| | **OVX n = 10, N = 5** | | **OVX n = 11, N = 4** |
| *Figure 1a, b* | **OVX + E n = 10, N = 6** | | **OVX + E n = 9, N = 5** |
| *Figure 1c–f,* *Figure 1—figure supplement 1a* left, 1b left | Control | | KERKO |
| | Intact n = 11, N = 4 | | Intact n = 11, N = 5 |
| | OVX n = 11, N = 5 | | OVX n = 9, N = 4 |
| | OVX + E n = 11, N = 7 | | OVX + E n = 12, N = 5 |
| *Figure 2* | Control | | KERKO |
| | n = 8, N = 4 | | n = 7, N = 4 |
| *Figure 3d,e* | AVPV-AAV-*lacZ* | AVPV-AAV-*Esr1*g1 | AVPV-AAV-*Esr1*g2 |
| | N = 3 | N = 3 | N = 4 |
| *Figure 3f* | AVPV-AAV- *lacZ* | | AVPV-AAV-*Esr1* |
| | N = 6 | | N = 8 (g1 N = 4, g2 N = 4) |
| *Figure 3g* | AVPV-AAV- *lacZ* | | AVPV-AAV-*Esr1* |
| | N = 6 | | N = 9 (g1 N = 5, g2 N = 4) |
| *Figure 4d–j* and *Figure 1—figure supplement 1a* middle, 1b middle | IF *post hoc* | | PCR *post hoc* |
| | *Esr1* n = 15, N = 5 | | *Esr1* n = 10, N = 4 |
| | *lacZ* n = 14, N = 4 | | *lacZ* n = 9, N = 3 |
| | uninfected n = 8, N = 4 | | uninfected n = 4, N = 2 |
| *Figure 5a–d* | Arc-AAV-*lacZ* | Arc-AAV-*Esr1*g1 | Arc-AAV-*Esr1*g2 |
| | N = 6 | N = 4 | N = 4 |
| *Figure 5e–g* | Arc-AAV-*lacZ* | | Arc-AAV-*Esr1* |
| | N = 6 | | N = 8 (g1 N = 4, g2 N = 4) |
| *Figure 6a–c* | Arc-AAV-*lacZ* | | Arc-AAV-*Esr1* |
| | n = 11, N = 5 | | n = 13, N = 5 |
| *Figure 6d–f* | Arc-AAV-*lacZ* | | Arc-AAV- *Esr1* |
| | n = 10, N = 5 | | n = 12, N = 5 |
| *Figure 1—figure supplement 1a* left, 1b right | KERKO | | AVPV-AAV-*Esr1* |
| | n = 12, N = 5 | | n = 25, N = 9 |
| *Figure 1—figure supplement 1c* left | Control | | KERKO |
| | Intact N = 6 | | Intact n = 11, N = 7 |
| | OVX N = 6 | | OVX n = 11, N = 6 |
| | OVX + E N = 5 | | OVX + E n = 9, N = 7 |
| *Figure 1—figure supplement 1c* middle | AVPV-AAV-*lacZ* | | AVPV-AAV-*Esr1* |
| | N = 7 | | N = 9 |
| *Figure 1—figure supplement 1c* middle | Arc-AAV-*lacZ* | | Arc-AAV-*Esr1* |
| | N = 5 | | N = 5 |
| *Figure 4—figure supplement 1* | AVPV-AAV-*lacZ* | | AVPV-AAV-*Esr1* |
| | n = 16, N = 5 | | n = 23, N = 5 (g1 N = 3, g2 N = 2) |

DOI: https://doi.org/10.7554/eLife.43999.018

animals. For IF staining, LH surge and LH pulse measurements, and reproductive cyclicity, at least three mice were tested per AAV vector.

## Acknowledgements

We thank Elizabeth Wagenmaker for expert technical assistance. We thank Dr. Daniel Michele for sharing C2C12 cell lines with us. We thank University of Virginia Ligand Core for performing the ultra-sensitive LH assay. We thank the University of Virginia Center for Research in Reproduction Ligand Assay and Analysis Core for LH assays. The Core is supported by the Eunice Kennedy Shriver NICHD/NIH (NCTRI) Grant P50-HD28934 to John C. Marshall.

## Additional information

### Funding

| Funder | Grant reference number | Author |
|---|---|---|
| National Institute of Diabetes and Digestive and Kidney Diseases | R01 DK095201 | Yatrik M Shah |
| National Institute of Diabetes and Digestive and Kidney Diseases | P30DK020572 (Michigan Diabetes Research Center) | Martin G Myers |
| Eunice Kennedy Shriver National Institute of Child Health and Human Development | R01 HD41469 | Suzanne M Moenter |

The funders had no role in study design, data collection and interpretation, or the decision to submit the work for publication.

### Author contributions

Luhong Wang, Conceptualization, Resources, Data curation, Software, Formal analysis, Funding acquisition, Validation, Visualization, Methodology, Writing—original draft, Project administration, Writing—review and editing; Charlotte Vanacker, Data curation, Formal analysis, Visualization, Writing—review and editing; Laura L Burger, Formal analysis, Methodology, Writing—review and editing; Tammy Barnes, Data curation, Methodology; Yatrik M Shah, Resources, Methodology, Writing—review and editing; Martin G Myers, Conceptualization, Resources, Methodology, Writing—review and editing; Suzanne M Moenter, Conceptualization, Resources, Data curation, Software, Formal analysis, Supervision, Funding acquisition, Validation, Investigation, Visualization, Methodology, Writing—original draft, Project administration, Writing—review and editing

### Author ORCIDs

Luhong Wang [iD] http://orcid.org/0000-0002-1085-841X
Charlotte Vanacker [iD] http://orcid.org/0000-0001-5289-9298
Laura L Burger [iD] http://orcid.org/0000-0002-9545-0287
Tammy Barnes [iD] http://orcid.org/0000-0003-2566-7445
Martin G Myers [iD] http://orcid.org/0000-0001-9468-2046
Suzanne M Moenter [iD] http://orcid.org/0000-0001-9812-0497

### Ethics

Animal experimentation: This study was performed in strict accordance with the recommendations in the Guide for the Care and Use of Laboratory Animals of the National Institutes of Health. All of the animals were handled according to approved institutional animal care and use committee (IACUC) protocols (Pro 000006816 and Pro000008797) of the University of Michigan. The protocol was approved by the University of Michigan Institutional Animal Care and Use Committee. Every effort was made to minimize suffering.

Decision letter and Author response

Decision letter https://doi.org/10.7554/eLife.43999.021
Author response https://doi.org/10.7554/eLife.43999.022

## Additional files

### Supplementary files

• Transparent reporting form
DOI: https://doi.org/10.7554/eLife.43999.019

### Data availability

No new dataset is generated or used in the current study.

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
