## [Decision Letter]

Thank you for submitting your article "Genetic dissection of the different roles of hypothalamic kisspeptin neurons in regulating female reproduction" for consideration by *eLife*. Your article has been reviewed by three peer reviewers, and the evaluation has been overseen by a Reviewing Editor and Catherine Dulac as the Senior Editor. The following individual involved in review of your submission has agreed to reveal their identity: Vincent Prevot (Reviewer #3).

The reviewers have discussed the reviews with one another and the Reviewing Editor has drafted this decision to help you prepare a revised submission.

Summary:

Wang and colleagues describe results from a series of studies aimed at identifying the physiological roles of estrogen receptors in distinct populations of kisspeptin neurons in the hypothalamus. Using multidisciplinary approaches, the deleted ERα in kisspeptin neurons postnatally. This is an important advance in the field as previous mouse models assessing the functional role of ERα in kisspeptin neurons were neither spatially nor temporally restricted therefore revealing important roles during development (such as in puberty onset) but leaving the role in adult reproductive physiology somewhat unexplored. The findings address a long-standing question, which is central to the field of reproductive neuroendocrinology. Fifteen years after the discovery of the two populations of hypothalamic neurons expressing kisspeptin, which is a neuropeptide playing a key role in puberty onset and fertility, the authors use a conditional and time-controlled Crispr-Cas9 approach in adult mice to demonstrate that estrogen receptor (ER) α expression in the anteroventral periventricular (AVPV) population of kisspeptin neurons is required for the onset of the preovulatory surge and that its expression in the population of kisspeptin neurons in the arcuate nucleus of the hypothalamus (ARH) is required for estrous cyclicity and plays a role in the control of pulsatile LH release. Collectively, the studies are well described and provide novel observations that will be of wide interest. Several technical issues need to be addressed.

Essential revisions:

The microinjection procedures need some further explanation including assessments and definition of "hits" and misses. Also, the number of mice in each group is a bit difficult to understand.

Also, the authors should comment a bit more about the predicted "off target" effects of the sgRNAs being used.

I am not fully convinced that the deletion of ERα in AVPV kisspeptin neurons in adulthood does not affect estrous cycles. While it is principally more difficult if not impossible to substantiate a negative finding experimentally, in this case it is even more difficult as only a good half of the estrogen-sensitive AVPV kisspeptin neurons are affected by the experimental paradigm presented here (the authors report that 28 ± 1% of AVPV kisspeptin cells expressed ERα post AAV infection while 72 ± 2% of AVPV kisspeptin neurons expressed ERα in the controls; subsection “Design and validation of sgRNAs that target *Esr1*”). Instead of postulating other cells expressing ERα in the control of cycylicity (Discussion, third paragraph), wouldn't the intact other half of the estrogen-sensitive AVPV kisspeptin neurons that still express ERα after AAV injection be the simplest explanation for this? This should be discussed.

The images in Figure 3 and in Figure 3—figure supplement 1 and Figure 5—figure supplement 1 are so small that I cannot distinguish any cellular details.

Intriguingly, deletion of the ERα in AVPV Kispeptin neurons blunts the preovulatry surge but does not appear to affect estrous cyclicity. Did the authors check ovarian cytology in their mice to know whether the animals actually ovulate? Do they have any data on the cyclicity of estrogen levels? Did they also check LH pulsatility in these mice?

---

## [Author Response]

Essential revisions:The microinjection procedures need some further explanation including assessments and definition of "hits" and misses. Also, the number of mice in each group is a bit difficult to understand.

Definition of how hits and misses were evaluated are now presented in the Materials and methods, and new Table 5 has been included to clarify the number of cells and mice for each group.

Also, the authors should comment a bit more about the predicted "off target" effects of the sgRNAs being used.

This is a good point and we have included the sites predicted by the Feng Zhang’s guide design tool (http://crispr.mit.edu) as possible off-target binding sites are listed in new Table 3. This software is currently not available but is what was used at the start of these studies several years ago. We thus confirmed the sites by software Benchling (https://benchling.com/academic). Benchling predicted fewer sites compared to the previous analysis, but provided no new sites. We included the results from Feng Zhang’s guide design tool in the manuscript as these appear to be more rigorous.

The images in Figure 3 and in Figures 3—figure supplement 1 and Figure 5—figure supplement 1 are so small that I cannot distinguish any cellular details.

The goal of the supplementary figures is to show the general region infected rather than cellular detail, thus these have not been altered. Figure 3D has been enlarged as requested.

I am not fully convinced that the deletion of ERα in AVPV kisspeptin neurons in adulthood does not affect estrous cycles. While it is principally more difficult if not impossible to substantiate a negative finding experimentally, in this case it is even more difficult as only a good half of the estrogen-sensitive AVPV kisspeptin neurons are affected by the experimental paradigm presented here (the authors report that 28 ± 1% of AVPV kisspeptin cells expressed ERα post AAV infection while 72 ± 2% of AVPV kisspeptin neurons expressed ERα in the controls; subsection “Design and validation of sgRNAs that target Esr1”). Instead of postulating other cells expressing ERα in the control of cycylicity (Discussion, third paragraph), wouldn't the intact other half of the estrogen-sensitive AVPV kisspeptin neurons that still express ERα after AAV injection be the simplest explanation for this? This should be discussed.Intriguingly, deletion of the ERα in AVPV Kispeptin neurons blunts the preovulatry surge but does not appear to affect estrous cyclicity. Did the authors check ovarian cytology in their mice to know whether the animals actually ovulate? Do they have any data on the cyclicity of estrogen levels? Did they also check LH pulsatility in these mice?

The reviewers make a good point with these two comments, which we will address together. The viral CRISPR approach produced a knock down, not a knock out. It is thus possible that the remaining cells with ERα in the AVPV support the continuation of the cycle and we have included this in the Discussion. We think it is the weight of the cycle and surge observations together that need to be considered. LH surge amplitude is clearly attenuated in these mice. This includes both the proestrous surge during the cycle, and the estradiol-induced surge. We included the latter as we shared the reviewer’s concern that the cycle may not produce an adequate estradiol level, but this concern is alleviated by providing a sufficient estradiol signal is provided via implantation. Because cycles continue and vaginal cornification depends upon an estradiol rise, we can conclude some cyclic variation of estradiol occurs. In two mice injected with virus in this region, we measured uterine mass on the afternoon of proestrus and it was well within the range that is typically observed in control mice. Uterine mass is linearly related to estradiol levels (Endocrinology 141:396). Vaginal cornification, limited uterine mass data and failure to exhibit a typical LH surge in response to exogenous estradiol together point to a likely sufficient estradiol signal, thus a disruption of the surge-generating mechanism. We cannot, however, rule out a more limited estradiol rise contributing to the blunting of the proestrous surge and have added this to the Discussion. We did not assess LH pulses in these mice. Ovarian cytology in rodents is complicated by the lifespan of the corpra lutea extending several cycles. Ovulation points themselves are difficult to assess histologically. We did, however, examine serial sections through the ovary of these mice as well as some mice with arcuate-targeted viral injections. Mice with *lacZ* targeted to the AVPV had between 10 and 12 corpra lutea (CL) per mouse. There was variability in the *Esr1* mice, with two having similar numbers to controls (11 and 12 CL), two having only four CL each and two not showing any evidence of ovulation. These data have been added to the text of the results and suggest that ovulation is indeed impaired in at least some mice in which estrogen receptor α expression was reduced in AVPV kisspeptin neurons.